# ON THE DESIGN OF KL-REGULARIZED POLICY GRADIENT ALGORITHMS FOR LLM REASONING

**Yifan Zhang**[*,◇,1,2]  **Yifeng Liu**[*,1]  **Huizhuo Yuan**[1]  **Yang Yuan**
**Quanquan Gu**[1†]  **Andrew Chi-Chih Yao**[3†]

[1]University of California, Los Angeles   [2]Princeton University   [3]Tsinghua University

yifzhang@princeton.edu, liuyifeng@cs.ucla.edu
qgu@cs.ucla.edu, andrewcyao@tsinghua.edu.cn

## ABSTRACT

Policy gradient algorithms have been successfully applied to enhance the reasoning capabilities of large language models (LLMs). KL regularization is ubiquitous, yet the design surface, choice of KL direction (forward vs. reverse), normalization (normalized vs. unnormalized), and estimator ($k_1/k_2/k_3$), is scattered across the literature and often intertwined with off-policy estimation. We ask a focused question: under the off-policy setting, what weighting is required for each KL variant so that the surrogate we optimize yields the exact gradient of the intended KL-regularized objective? We answer this with a compact, unified derivation we call the Regularized Policy Gradient (**RPG**) view. RPG (i) unifies normalized and unnormalized KL variants and shows that the widely-used $k_3$ penalty is exactly the unnormalized KL; (ii) specifies conditions under which REINFORCE-style losses with stop-gradient are gradient-equivalent to fully differentiable surrogates; (iii) identifies and corrects an off-policy importance-weighting mismatch in GRPO's KL term; and (iv) introduces RPG-Style Clip, a clipped-importance-sampling step within RPG-REINFORCE that enables stable, off-policy policy-gradient training at scale. On mathematical reasoning benchmarks (AIME24, AIME25), RPG-REINFORCE with RPG-Style Clip improves accuracy by up to +6 absolute percentage points over DAPO. We extend our experiments to 8K context length, and RPG-REINFORCE with RPG-Style Clip achieves 52% accuracy on AIME25, surpassing the official Qwen3-4B-Instruct model (47%). Notably, RPG is a stable and scalable RL algorithm for LLM reasoning, realized via (a) a KL-correct objective, (b) clipped importance sampling, and (c) an iterative reference-policy update scheme. Project Page: https://github.com/complex-reasoning/RPG.

## 1 INTRODUCTION

Reinforcement learning (RL), particularly policy gradient (PG) methods, provides a powerful framework for solving sequential decision-making problems in complex environments. These methods have been successfully applied in diverse domains, ranging from robotics to game playing, and have recently become instrumental in fine-tuning large language models (LLMs) to align with human preferences and instructions (Ouyang et al., 2022) and enhancing the reasoning capabilities of LLMs (Shao et al., 2024; Guo et al., 2025). Classical PG algorithms like REINFORCE (Williams, 1992) optimize policies directly but often suffer from high gradient variance. Advanced methods like Proximal Policy Optimization (PPO) (Schulman et al., 2017) improve stability and sample efficiency, enabling large-scale applications, often by operating in an off-policy manner and employing techniques like training critic models for the estimation of value functions. Our theme in this paper is stability and scalability: which design choices in KL-regularized PG matter for robustness under off-policy sampling, and practical throughput on modern LLM stacks?

A crucial technique for stabilizing policy optimization, especially when deviating from strictly on-policy updates or aiming to control policy complexity, is regularization. Kullback-Leibler (KL) divergence is a commonly used regularizer, penalizing the deviation of the learned policy $\pi_\theta$ from

---

*Equal contribution; †Corresponding authors; ◇Work was done during a visit to UCLA.

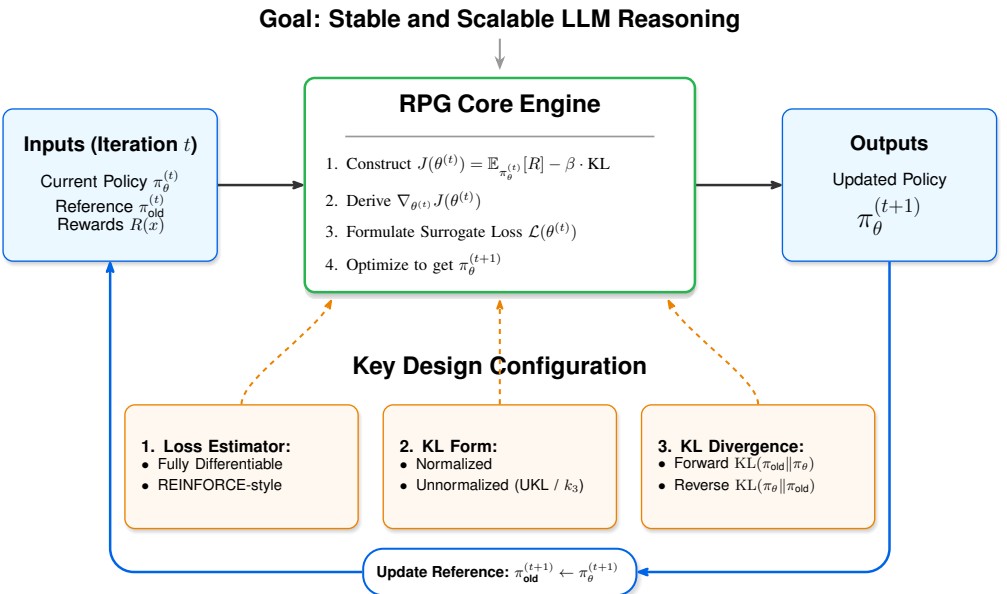

Figure 1: Overview of the iterative Regularized Policy Gradient (RPG) framework proposed in this work. At each iteration $t$, the central RPG Core Engine processes inputs: the current policy $\pi_\theta^{(t)}$, a reference policy $\pi_{\text{old}}^{(t)}$, and associated rewards $R(x)$. The engine's operation encompasses four main steps: (1) constructing the KL-regularized objective $J(\theta^{(t)})$, which combines the expected reward with a KL divergence term; (2) deriving the off-policy policy gradient $\nabla_{\theta^{(t)}} J(\theta^{(t)})$; (3) formulating a corresponding surrogate loss function $\mathcal{L}(\theta^{(t)})$; and (4) optimizing the policy parameters to yield an updated policy $\pi_\theta^{(t+1)}$, aimed at enhancing LLM reasoning capabilities. The specific behavior of the RPG Core Engine is configured by three key design choices: (i) the Loss Estimator type (Fully Differentiable or REINFORCE-style with Stop-Gradient); (ii) the KL Form (Normalized or Un-normalized, e.g., using UKL / $k_3$ estimators); and (iii) the KL Divergence Type (Forward $\text{KL}(\pi_{\text{old}}\|\pi_\theta)$ or Reverse $\text{KL}(\pi_\theta\|\pi_{\text{old}})$). The framework operates iteratively, with the updated policy $\pi_\theta^{(t+1)}$ from one iteration informing the inputs for the next, including the update of the reference policy $\pi_{\text{old}}^{(t+1)}$, to facilitate continuous learning and performance improvement.

a reference policy $\pi_{\text{ref}}$ (e.g., policy from previous iteration $\pi_{\theta_{\text{old}}}$ or a fixed prior policy $\pi^{\text{SFT}}$). KL regularization helps prevent destructive policy updates, encourages exploration around known good policies, and can prevent catastrophic forgetting or overly confident outputs (Ouyang et al., 2022).

Despite the widespread use of KL regularization in methods such as PPO (often implicitly through reward penalties) and explicit formulations like GRPO (Shao et al., 2024), there exists a considerable variety in how the KL divergence is formulated and estimated. Different choices include Forward KL and Reverse KL, handling potentially unnormalized distributions (Minka et al., 2005) (leading to unnormalized KL (UKL) and unnormalized reverse KL (URKL) formulations), and the use of various estimators like the $k_2$ and $k_3$ estimators (Schulman, 2020) designed to potentially reduce variance or offer different properties compared to the standard log-ratio ($k_1$) estimator. Furthermore, the interplay between the choice of KL formulation, the policy optimization setting (on-policy vs. off-policy), and the derivation of appropriate surrogate loss functions (fully differentiable vs. REINFORCE-style gradient estimators) can lead to subtle differences.

This paper provides systematic derivations and a unifying treatment of KL-regularized policy gradient methods, and revisits classical REINFORCE through the lens of clipped importance sampling. Our main contributions are summarized as follows:

- We derive policy gradients and corresponding surrogate losses for Forward/Reverse KL, in normalized (KL) and unnormalized (UKL) forms, under off-policy sampling with importance weights.

- We give both fully differentiable surrogates and REINFORCE-style losses (with stop-gradient) and prove their gradient-equivalence to the intended regularized objective (Proposition 4.1, Appendix L).
- We introduce RPG-Style Clip, a clipped-importance-weighted REINFORCE estimator that substantially improves stability and variance control while preserving the RPG gradients.
- We reveal the equality between the $k_3$ estimator and unnormalized KL (Appendix D), and show that GRPO's KL penalty omits an essential importance weight under off-policy sampling. We provide a corrected estimator and loss consistent with the intended objective.
- We present an iterative training framework that periodically updates the reference model to satisfy KL constraints while allowing the policy to depart meaningfully from the initial checkpoint.
- On math reasoning, RPG-REINFORCE (with RPG-Style Clip) yields stable and scalable training and outperforms DAPO by up to +6 absolute points on AIME24/25.
- We extend our experiments to 8K context length and find that RPG-REINFORCE with RPG-Style Clip achieves 52% accuracy on AIME25, surpassing the official Qwen3-4B-Instruct model (47%) and outperforming strong baselines.

## 2 PRELIMINARIES

Policy gradient (PG) methods are a cornerstone of modern reinforcement learning (RL), optimizing parameterized policies $\pi_\theta$ by estimating the gradient of an expected objective function $J(\theta)$ with respect to the policy parameters $\theta$. Typically, $J(\theta)$ represents the expected cumulative discounted reward over trajectories $\tau = (s_0, a_0, r_0, s_1, \ldots, s_T, a_T, r_T)$ generated by the policy: $J(\theta) = \mathbb{E}_{\tau \sim \pi_\theta}[G(\tau)]$, where $G(\tau) = \sum_{t=0}^{T} \gamma^t r_t$ is the trajectory return (with discount factor $\gamma$), and the expectation is taken over the trajectories sampled according to the policy $\pi_\theta(a|s)$ and the environment dynamics $p(s'|s, a)$. The Generalized Policy Gradient Theorem (GPPT) provides a foundation for deriving these gradients (see Appendix J for the proof).

**Proposition 2.1** (Generalized Policy Gradient Theorem). *Let $\pi_\theta(x)$ be a probability density or mass function parameterized by $\theta$, representing the probability of sampling item $x$. Let $f(x, \theta)$ be a scalar-valued function associated with $x$, potentially depending on $\theta$. Under suitable regularity conditions, the gradient of the expectation $\mathbb{E}_{x \sim \pi_\theta}[f(x, \theta)]$ with respect to $\theta$ is:*

$$\nabla_\theta \mathbb{E}_{x \sim \pi_\theta}[f(x, \theta)] = \mathbb{E}_{x \sim \pi_\theta}\left[f(x, \theta)\nabla_\theta \log \pi_\theta(x) + \nabla_\theta f(x, \theta)\right]. \tag{2.1}$$

The term $\mathbb{E}[f\nabla \log \pi]$ reflects how changes in $\theta$ affect the probability of sampling $x$, while $\mathbb{E}[\nabla f]$ reflects how changes in $\theta$ directly affect the function value $f$.

The classic REINFORCE algorithm (Williams, 1992) applies the GPPT to the standard RL objective $J(\theta) = \mathbb{E}_{\tau \sim \pi_\theta}[G(\tau)]$. In this case, $f(\tau, \theta) = G(\tau)$, the total trajectory return, which does not depend directly on $\theta$ (i.e., $\nabla_\theta G(\tau) = 0$). The theorem simplifies, and the gradient can be expressed using per-timestep contributions (Sutton et al., 1998):

$$\nabla_\theta J(\theta) = \mathbb{E}_{\tau \sim \pi_\theta}\left[\sum_{t=0}^{T} G_t \nabla_\theta \log \pi_\theta(a_t|s_t)\right],$$

where $G_t = \sum_{k=t}^{T} \gamma^{k-t} r_k$ is the return-to-go from timestep $t$. Due to space limit, we defer the detailed introduction of REINFORCE to Appendix C.1.

### 2.1 KL REGULARIZATION IN POLICY GRADIENTS

A common technique to stabilize policy optimization, especially in off-policy settings or when fine-tuning large models, is regularization. The Kullback-Leibler (KL) divergence is frequently used to penalize the deviation of the learned policy $\pi_\theta$ from a reference policy $\pi_{\text{ref}}$ (which could be $\pi_{\theta_{\text{old}}}$, an initial supervised fine-tuned model, or another prior). $\text{KL}(P \| Q) \geq 0$ with equality iff $P = Q$ almost everywhere. It is asymmetric (i.e., $\text{KL}(P \| Q) \neq \text{KL}(Q \| P)$). Minimizing the forward KL $\text{KL}(\pi_{\text{ref}} \| \pi_\theta)$ encourages $\pi_\theta$ to cover the support of $\pi_{\text{ref}}$ (zero-forcing), while minimizing the reverse KL $\text{KL}(\pi_\theta \| \pi_{\text{ref}})$ encourages $\pi_\theta$ to be concentrated where $\pi_{\text{ref}}$ has high probability mass (mode-seeking).

Adding a KL penalty to the RL objective, such as $J(\theta) = \mathbb{E}_{\pi_\theta}[R] - \beta \, \text{KL}(\pi_\theta \| \pi_{\text{ref}})$, helps control the policy update size, prevents large deviations from $\pi_{\text{ref}}$, encourages exploration near known good

policies, and can mitigate issues like catastrophic forgetting or overly confident outputs, particularly relevant in LLM fine-tuning (Ouyang et al., 2022). For PPO (see Appendix C.2), this penalty can be incorporated implicitly via reward shaping: $r'_t = r_t - \beta \log(\pi_\theta(a_t|s_t)/\pi_{\text{ref}}(a_t|s_t))$. Alternatively, it can be added explicitly to the objective function, as in GRPO. The specific form of the KL divergence (forward/reverse), whether distributions are normalized (KL vs. UKL), and the choice of estimator (e.g., standard log-ratio vs. $k_3$ estimator (Schulman, 2020)) can vary, leading to different properties (mode seeking v.s. zero-forcing) and gradient estimators, as explored later in this paper (Sections 3 and 4).

## 2.2 GROUP RELATIVE POLICY OPTIMIZATION (GRPO)

Group Relative Policy Optimization (GRPO) (Shao et al., 2024) adapts the PPO framework for training LLMs, notably by eliminating the need for a learned value function (critic). Instead of using GAE, GRPO estimates the advantage $\widehat{A}_{i,t}$ at token $t$ of output $o_i$ based on the relative rewards within a group of $G$ outputs $\{o_1, \ldots, o_G\}$ sampled from the old policy $\pi_{\theta_{\text{old}}}$ for the same prompt $q$.

Crucially, GRPO modifies the PPO objective by explicitly adding a KL regularization term directly to the objective function. Its objective (simplified notation) is:

$$\mathcal{J}_{\text{GRPO}}(\theta) = \mathbb{E}_{q \sim P(Q), \{o_i\} \sim \pi_{\text{old}}} \left[ \frac{1}{G} \sum_{i=1}^{G} \frac{1}{|o_i|} \sum_{t=1}^{|o_i|} \left( J_{i,t}^{\text{Clip}}(\theta) - \beta \cdot \text{KL}_{\text{est}}\big(\pi_\theta(\cdot|h_{i,t}) \| \pi_{\text{ref}}(\cdot|h_{i,t})\big) \right) \right],$$

where $h_{i,t} = (q, o_{i,<t})$ is the history, $J_{i,t}^{\text{Clip}}(\theta)$ represents the PPO-Clip term from Eq. (C.3) applied using the group-relative advantage estimate $\widehat{A}_{i,t}$, and $\pi_{\text{ref}}$ is a reference model (e.g., the initial SFT model). For the KL penalty, GRPO employs the $k_3$ estimator form (Schulman, 2020), evaluated per token $o_{i,t}$:

$$\text{KL}_{\text{est}}(\pi_\theta \| \pi_{\text{ref}}) \approx k_3 \left( \frac{\pi_{\text{ref}}(o_{i,t} \mid h_{i,t})}{\pi_\theta(o_{i,t} \mid h_{i,t})} \right) = \frac{\pi_{\text{ref}}(o_{i,t} \mid h_{i,t})}{\pi_\theta(o_{i,t} \mid h_{i,t})} - \log \frac{\pi_{\text{ref}}(o_{i,t} \mid h_{i,t})}{\pi_\theta(o_{i,t} \mid h_{i,t})} - 1.$$

This uses the functional form $k_3(y) = y - \log y - 1$ as discussed in Schulman (2020), applied with $y = \pi_{\text{ref}}(o_{i,t}|h_{i,t})/\pi_\theta(o_{i,t}|h_{i,t})$. This form is related to the unnormalized reverse KL divergence, $\text{UKL}(\pi_\theta \| \pi_{\text{ref}})$ (see Section 3.2 and Appendix D for a detailed discussion). However, a key observation regarding GRPO's KL penalty is its estimation. If the KL penalty in GRPO is intended to approximate $\beta \cdot \text{UKL}(\pi_\theta(\cdot|h_{i,t}) \| \pi_{\text{ref}}(\cdot|h_{i,t}))$, its off-policy estimation (sampling $o_{i,t}$ from $\pi_{\text{old}}$) would generally involve an importance weight $w_{i,t} = \frac{\pi_\theta(o_{i,t}|h_{i,t})}{\pi_{\text{old}}(o_{i,t}|h_{i,t})}$ multiplying the $k_3$ term. The direct subtraction without this weight means the gradient derived from GRPO's objective does not, in general, correspond to the gradient of the intended off-policy objective $J^{\text{Clip}} - \beta \, \text{UKL}(\pi_\theta \| \pi_{\text{ref}})$. For clarity, a corrected off-policy estimator for the GRPO KL component at history $h_{i,t}$ is

$$\widehat{\text{KL}}_{\text{GRPO-corrected}}(h_{i,t}; \theta) = \mathbb{E}_{o_{i,t} \sim \pi_{\text{old}}(\cdot|h_{i,t})} \left[ w_{i,t} \, k_3 \left( \frac{\pi_{\text{ref}}(o_{i,t}|h_{i,t})}{\pi_\theta(o_{i,t}|h_{i,t})} \right) \right],$$

which is consistent with URKL/UKL depending on direction (see Section 3 and Appendix D). Our results in Section 3 provide derivations for KL-regularized objectives that explicitly account for off-policy sampling via importance weights. Related work is detailed in Appendix A.

## 3 REGULARIZED POLICY GRADIENTS

In this section, we start from the KL regularized objective $J(\theta) = \mathbb{E}[R] - \beta \, \text{KL}$ and we treat this as the exact target for training. Then we derive its true gradient under off-policy sampling. The derivation shows that we need precise importance weighting so that the gradient of the surrogate loss matches the gradient of this objective. The weights are different for Forward vs. Reverse KL, as summarized in Table 1. This viewpoint unifies many existing estimators within a single framework and clarifies why the KL term in GRPO can lead to unstable updates when its weighting is chosen improperly. In the main text, we focus on the unnormalized objectives (UFKL/URKL), while the normalized formulations (FKL/RKL) and their losses are deferred to Appendix E (see also Table 4). All proofs are provided in Appendix K.

## 3.1 UNNORMALIZED FORWARD KL REGULARIZATION

In scenarios where distributions might not be normalized (i.e., $\int_x \pi(x)dx \neq 1$), the standard KL divergence might not fully capture the dissimilarity. The unnormalized forward KL divergence addresses this by adding a mass correction term. Let $\pi_{\text{old}}(x)$ be a potentially unnormalized reference measure with total mass $Z_{\text{old}} = \int_x \pi_{\text{old}}(x)dx$. Let $\widetilde{\pi}_{\text{old}}(x) = \pi_{\text{old}}(x)/Z_{\text{old}}$ be the corresponding normalized probability distribution, such that $\int \widetilde{\pi}_{\text{old}}(x)dx = 1$.

Table 1: Summary of fully differentiable surrogate loss functions $\mathcal{L}(\theta)$ for unnormalized KL-regularized objectives (main text). Minimizing $\mathcal{L}(\theta)$ corresponds to maximizing $J(\theta) = \mathbb{E}_{\pi_\theta}[R(x)] - \beta \cdot \text{Divergence}$. Samples $x$ are drawn from $\widetilde{\pi}_{\text{old}} = \pi_{\text{old}}/Z_{\text{old}}$. These losses yield $-\nabla_\theta J(\theta)$ via differentiation. Notation: $w(x) = \pi_\theta(x)/\pi_{\text{old}}(x)$, $R(x)$ is reward, $\beta$ the regularization strength, and $Z_{\text{old}} = \int \pi_{\text{old}}$. Normalized counterparts are in Appendix E (Table 4).

| Regularization (Unnormalized) | Surrogate loss (expectation w.r.t. $\widetilde{\pi}_{\text{old}}$) |
|---|---|
| **Forward (UFKL)** | $Z_{\text{old}}\, \mathbb{E}\big[-w(x)R(x) + \beta\big(w(x) - \log w(x) - 1\big)\big]$ |
| **Reverse (URKL)** | $Z_{\text{old}}\, \mathbb{E}\big[-w(x)R(x) + \beta\big(w(x)\log w(x) - w(x)\big)\big]$ |

**Definition 3.1** (Unnormalized Forward KL). The unnormalized forward KL divergence (Minka et al., 2005; Zhu and Rohwer, 1995) between the measure $\pi_{\text{old}}$ and the density $\pi_\theta$ is defined as:

$$\text{UKL}(\pi_{\text{old}}\|\pi_\theta) = \underbrace{\int_x \pi_{\text{old}}(x)\log\frac{\pi_{\text{old}}(x)}{\pi_\theta(x)}\,dx}_{\text{Generalized KL}} + \underbrace{\int_x \big(\pi_\theta(x) - \pi_{\text{old}}(x)\big)\,dx}_{\text{Mass Correction}}.$$

This form is particularly relevant when dealing with reference measures that may not be perfectly normalized or when connecting to certain KL estimators like $k_3$ (see Remark 3.5).

Consider the objective using UKL regularization as follows:

$$J_{\text{UFKL}}(\theta) = \mathbb{E}_{x \sim \pi_\theta}[R(x)] - \beta\; \text{UKL}(\pi_{\text{old}}\|\pi_\theta). \tag{3.1}$$

To estimate this off-policy using samples from the normalized reference $\widetilde{\pi}_{\text{old}}(x) = \pi_{\text{old}}(x)/Z_{\text{old}}$, we define the importance weight $w(x) = \pi_\theta(x)/\pi_{\text{old}}(x)$ (using the unnormalized $\pi_{\text{old}}$). The gradient and corresponding loss function, incorporating the total mass $Z_{\text{old}}$ of the reference measure, are given in Proposition 3.2.

**Proposition 3.2** (Policy Gradient and Differentiable Loss for Unnormalized Forward KL). Consider the unnormalized KL regularized objective function in Eq. (3.1). The gradient of $J_{\text{UFKL}}(\theta)$ is:

$$\nabla_\theta J_{\text{UFKL}}(\theta) = Z_{\text{old}}\mathbb{E}_{x \sim \widetilde{\pi}_{\text{old}}}\left[\Big(w(x)R(x) - \beta\left(w(x) - 1\right)\Big)\nabla_\theta \log \pi_\theta(x)\right].$$

The corresponding surrogate loss for gradient descent optimization, estimated using samples $\{x_i\} \sim \widetilde{\pi}_{\text{old}}$, is:

$$\mathcal{L}_{\text{UFKL}}(\theta) = Z_{\text{old}}\mathbb{E}_{x \sim \widetilde{\pi}_{\text{old}}}\left[-w(x)R(x) + \beta\big(w(x) - \log w(x) - 1\big)\right],$$

satisfying $\nabla_\theta \mathcal{L}_{\text{UFKL}}(\theta) = -\nabla_\theta J_{\text{UFKL}}(\theta)$.

**Remark 3.3** (Interpretation of UFKL Loss and Gradient). The regularization component of the surrogate loss $\mathcal{L}_{\text{UFKL}}(\theta)$, specifically $Z_{\text{old}}\mathbb{E}_{x \sim \widetilde{\pi}_{\text{old}}}[\beta(w(x) - \log w(x) - 1)]$, corresponds to an off-policy estimate of the unnormalized forward KL divergence term $\beta \cdot \text{UKL}(\pi_{\text{old}}\|\pi_\theta)$ present in the objective $J_{\text{UFKL}}(\theta)$. This connection is established via the $k_3$ estimator (see Remark 3.5 and Appendix D). Furthermore, the gradient term $-\beta(w(x) - 1)$ effectively modifies the reward, guiding $\pi_\theta$ to match not only the shape of $\pi_{\text{old}}$ but also its overall mass $Z_{\text{old}}$, due to the mass correction component in $\text{UKL}(\pi_{\text{old}}\|\pi_\theta)$.

## 3.2 UNNORMALIZED REVERSE KL REGULARIZATION

Similar to the forward case, we can define an unnormalized reverse KL divergence, relaxing the normalization constraint on the reference distribution $\pi_{\text{old}}$. Let $\pi_{\text{old}}(x)$ be a potentially unnormalized reference measure with total mass $Z_{\text{old}} = \int \pi_{\text{old}}(x)dx$. Let $\widetilde{\pi}_{\text{old}}(x) = \pi_{\text{old}}(x)/Z_{\text{old}}$ be the corresponding normalized probability distribution.

**Definition 3.4** (Unnormalized Reverse KL). The unnormalized reverse KL divergence between the density $\pi_\theta$ and the measure $\pi_{\text{old}}$ is defined as:

$$\text{UKL}(\pi_\theta \| \pi_{\text{old}}) = \underbrace{\int_x \pi_\theta(x) \log \frac{\pi_\theta(x)}{\pi_{\text{old}}(x)}\, dx}_{\text{Generalized KL}} + \underbrace{\int_x \Big(\pi_{\text{old}}(x) - \pi_\theta(x)\Big) dx}_{\text{Mass Correction}}.$$

The mass correction term simplifies to $Z_{\text{old}} - \int \pi_\theta(x) dx$.

**Remark 3.5.** (Equivalence to $k_3$ estimator) The $k_3$ estimator (Schulman, 2020), often used for its empirical properties (e.g., in GRPO (Shao et al., 2024)), is defined for a density ratio $y(x)$ as:

$$k_3(y) := y - 1 - \log y. \tag{3.2}$$

As shown in Appendix D, this functional form directly relates to unnormalized KL divergences. For instance, $\text{KL}_{k_3}(\pi_\theta \| \pi_{\text{old}}) := \mathbb{E}_{x \sim \pi_\theta}[k_3(\pi_{\text{old}}(x)/\pi_\theta(x))]$ is equivalent to $\text{UKL}(\pi_\theta \| \pi_{\text{old}})$. This equivalence relationship justifies the exploration of UKL/URKL formulations within our framework.

Consider the objective using URKL:

$$J_{\text{URKL}}(\theta) = \mathbb{E}_{x \sim \pi_\theta}[R(x)] - \beta\ \text{UKL}(\pi_\theta \| \pi_{\text{old}}), \tag{3.3}$$

where UKL is defined above. As with UFKL, we derive the gradient and loss using expectations over the normalized reference $\widetilde{\pi}_{\text{old}}$ and the importance weight $w(x) = \pi_\theta(x)/\pi_{\text{old}}(x)$ (with unnormalized $\pi_{\text{old}}$). The results are summarized in Proposition 3.6.

**Proposition 3.6** (Policy Gradient and Differentiable Loss for Unnormalized Reverse KL). Consider the reverse unnormalized KL regularized objective function in Eq. (3.3). The gradient of $J_{\text{URKL}}(\theta)$ is:

$$\nabla_\theta J_{\text{URKL}}(\theta) = Z_{\text{old}} \mathbb{E}_{x \sim \widetilde{\pi}_{\text{old}}} \Big[ w(x) \Big( R(x) - \beta \log w(x) \Big) \nabla_\theta \log \pi_\theta(x) \Big].$$

A corresponding surrogate loss for gradient descent optimization, estimated using samples $\{x_i\} \sim \widetilde{\pi}_{\text{old}}$, is:

$$\mathcal{L}_{\text{URKL}}(\theta) = Z_{\text{old}} \mathbb{E}_{x \sim \widetilde{\pi}_{\text{old}}} \big[ -w(x)R(x) + \beta\big( w(x) \log w(x) - w(x) \big) \big],$$

satisfying $\nabla_\theta \mathcal{L}_{\text{URKL}}(\theta) = -\nabla_\theta J_{\text{URKL}}(\theta)$. The constant $Z_{\text{old}}$ scales the loss and gradient and may be omitted in practice.

**Remark 3.7** (URKL Loss and Mass Correction). The surrogate loss $\mathcal{L}_{\text{URKL}}(\theta)$ is designed such that its gradient is $-\nabla_\theta J_{\text{URKL}}(\theta)$. Specifically, the term $Z_{\text{old}} \mathbb{E}_{x \sim \widetilde{\pi}_{\text{old}}}[\beta(w(x) \log w(x) - w(x))]$ in the loss directly relates to the off-policy estimation of the unnormalized reverse KL divergence $\beta\,\text{UKL}(\pi_\theta \| \pi_{\text{old}})$, omitting a constant related to the total mass $Z_{\text{old}}$ which does not affect the gradient. The policy gradient's effective reward scaling factor, $(R(x) - \beta \log w(x))$, is simpler than its normalized RKL counterpart.

**Remark 3.8.** In Appendix B, we show the connection between RPG and the Natural Policy Gradient (NPG) (Kakade, 2001; Schulman et al., 2015). In particular, the NPG update is a special case of the RPG update, which uses a linear approximation for the expected return and a quadratic approximation for the KL regularization. This transforms the problem from simple first-order gradient ascent in PG (REINFORCE) into a second-order-like update: RPG.

### 3.3 RPG-STYLE CLIP: DUAL-CLIP TRUNCATION OF IMPORTANCE RATIOS

Large importance ratios $w(x) = \frac{\pi_\theta(x)}{\pi_{\text{old}}(x)}$ induce high variance and destabilize off-policy updates. Our **RPG-Style Clip** follows the dual-clip method implemented in Algorithm 1 in the Appendix: we clip $w$ into $[1 - \epsilon_1, 1 + \epsilon_2]$ and additionally impose a lower bound for negative advantages. Let $\widehat{A}(x; \theta)$ denote the regularized advantage analogue determined by the chosen objective (e.g., $\widehat{A}_{\text{URKL}} = (R - b) - \beta \log w$, $\widehat{A}_{\text{RKL}} = (R - b) - \beta(\log w + 1)$). The loss used in our implementation is

$$\mathcal{L}^{\text{RPG-Clip}}(x, \theta) = \begin{cases} \max\Big( -w(x)\,\widehat{A}(x; \theta),\ -\text{clip}(w(x), 1 - \epsilon_1, 1 + \epsilon_2)\,\widehat{A}(x; \theta) \Big), & \widehat{A}(x; \theta) \geq 0, \\ \min\Big( \max\big( -w(x)\,\widehat{A}(x; \theta),\ -\text{clip}(w(x), 1 - \epsilon_1, 1 + \epsilon_2)\,\widehat{A}(x; \theta) \big),\ -c\,\widehat{A}(x; \theta) \Big), & \widehat{A}(x; \theta) < 0, \end{cases} \text{ with}$$

$\epsilon_1, \epsilon_2 > 0$ and $c > 1$. The choice of $\widehat{A}$ for each divergence (URKL/UFKL/RKL/FKL) matches the gradients in Section 3 and is instantiated in Algorithm 1.

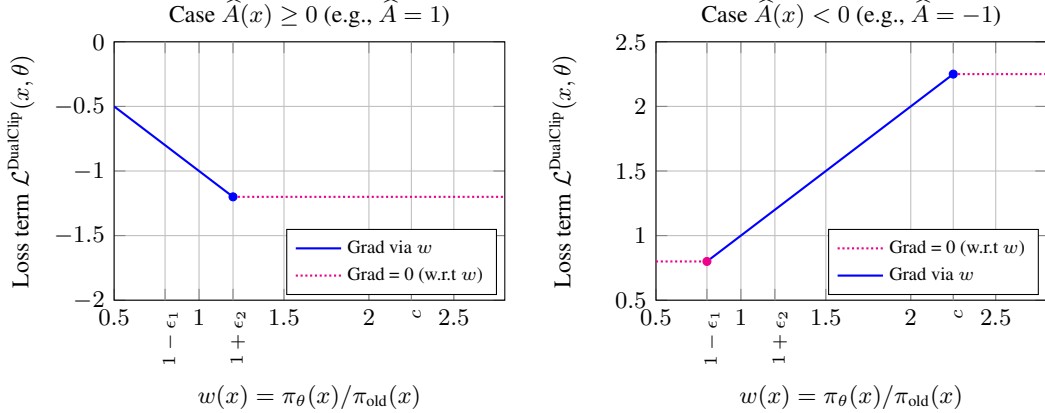

Figure 2: Visualization of the Dual-Clip loss term $\mathcal{L}^{\text{DualClip}}(x, \theta)$ vs. importance weight $w(x)$, as described in Section G.1 and Algorithm 1. This formulation is typically implemented as fully differentiable w.r.t $\theta$ (via $w(x)$ and potentially $\widehat{A}(x)$ if $\widehat{A}$ depends on $\theta$, e.g., via $\log w(x)$), unlike REINFORCE-style implementations that use $\text{SG}(\widehat{A})$ or $\text{SG}(\ell_i)$ within the loss. For visualization, $\widehat{A}(x)$ is treated as constant ($\widehat{A} = 1$ left, $\widehat{A} = -1$ right) to isolate the effect of $w$. **Solid blue:** Loss depends linearly on $w$, gradient $\nabla_\theta \mathcal{L}$ flows via $w(x)$. **Dotted magenta:** Loss is constant w.r.t $w$, gradient $\nabla_\theta \mathcal{L}$ does not flow via $w(x)$ in this segment (though it might flow via $\widehat{A}$ if $\widehat{A}$ depends on $\theta$). Left: Case $\widehat{A} < 0$. Right: Case $\widehat{A} \geq 0$.

## 4 REINFORCE-STYLE REGULARIZED POLICY GRADIENTS

Table 2: REINFORCE-style surrogate losses $\mathcal{L}(\theta)$ for unnormalized KL-regularized objectives using the stop-gradient operator (SG). These losses yield the target gradient via automatic differentiation. Compare with the fully differentiable losses in Table 1. Normalized versions are given in Appendix F.

| Regularization (Unnormalized) | REINFORCE-style loss (sampling $x \sim \widetilde{\pi}_{\text{old}}$) |
|---|---|
| **Forward (UFKL)** | $-\mathbb{E}\Big[ \text{SG}(Z_{\text{old}}(w(x)R(x) - \beta(w(x) - 1))) \log \pi_\theta(x) \Big]$ |
| **Reverse (URKL)** | $-\mathbb{E}\Big[ \text{SG}(Z_{\text{old}}w(x)(R(x) - \beta \log w(x))) \log \pi_\theta(x) \Big]$ |

In Section 3, we derived policy gradient estimators and corresponding fully differentiable surrogate losses $\mathcal{L}(\theta)$ for KL-regularized objectives. Those losses were constructed such that $\nabla_\theta \mathcal{L}(\theta) = -\nabla_\theta J(\theta)$ directly, typically by setting $\mathcal{L}(\theta) = -J_{\text{IS}}(\theta)$ (where $J_{\text{IS}}$ is the importance-sampled objective) up to constants. Notice that the gradients derived in Section 3 (Theorems 3.2 through 3.6) share a structural similarity with the REINFORCE estimator:

$$\nabla_\theta J(\theta) = \mathbb{E}_{x \sim \pi_{\text{sampling}}}\left[\text{Weight}(x, \theta)\nabla_\theta \log \pi_\theta(x)\right]$$

where $\pi_{\text{sampling}}$ is $\pi_{\text{old}}$ or its normalized version $\widetilde{\pi}_{\text{old}}$, and $\text{Weight}(x, \theta)$ encapsulates the reward and KL regularization terms, differing for each specific objective.

**Proposition 4.1** (Gradient-Equivalence of Surrogates). For each KL-regularized objective $J(\theta)$ derived in Section 3, the corresponding REINFORCE-style losses in Table 2 satisfy $\nabla_\theta \mathcal{L}(\theta) = -\nabla_\theta J(\theta)$ under the standard regularity assumptions used in the policy-gradient theorem. In particular, the stop-gradient operator ensures that dependence of the weight on $\theta$ (through importance ratios) does not leak unintended gradients. A proof sketch follows directly from the policy-gradient theorem and is completed in Appendix L.

This structural similarity motivates an alternative REINFORCE-style implementation using the stop-gradient operator SG. The general form of such losses and the detailed rationale for how they yield the target gradient via automatic differentiation are presented in Appendix F.1 (see Eq. (F.1)).

We explore these REINFORCE-style estimators as part of our framework, as they offer an alternative implementation path and demonstrate stronger or competitive empirical performance (Section 5). Proofs are in Appendix L. In the main text, we tabulate the unnormalized REINFORCE-style losses; normalized counterparts are deferred to Appendix F.

### 4.1 RPG-STYLE DUAL CLIP FOR REINFORCE-STYLE ESTIMATORS

Directly optimizing the REINFORCE-style losses in Table 2 can be unstable due to high variance in the importance weights $w(x)$. To mitigate this, we introduce **RPG-Style Clip**, which adapts the Dual-Clip technique (Ye et al., 2020) to our regularized setting.

The key strategy is to decompose the weight term inside the stop-gradient operator into the importance ratio $w(x)$ and a *regularized advantage* $\widehat{A}_{\text{reg}}(x)$. For example, for the URKL objective, we identify $\widehat{A}_{\text{reg}}(x) = Z_{\text{old}}(R(x) - \beta \log w(x))$ such that the weight is $w(x)\widehat{A}_{\text{reg}}(x)$. We then replace this linear weighting with a clipped version $\mathcal{C}(w(x), \text{SG}(\widehat{A}_{\text{reg}}(x)))$. The resulting loss is:

$$\mathcal{L}^{\text{RPG-Clip}}(\theta) = -\mathbb{E}_{x \sim \widetilde{\pi}_{\text{old}}} \left[ \mathcal{C}\left(w(x), \text{SG}(\widehat{A}_{\text{reg}}(x))\right) \log \pi_\theta(x) \right], \tag{4.1}$$

where $\mathcal{C}(w, A)$ applies dual-clip bounds, clipping $w$ to $[1 - \epsilon_1, 1 + \epsilon_2]$ for positive $A$, and enforcing a lower bound $c$ for negative $A$, as detailed in Appendix G.1. This ensures stable off-policy updates while preserving the correct gradient direction derived from the KL-regularized objective.

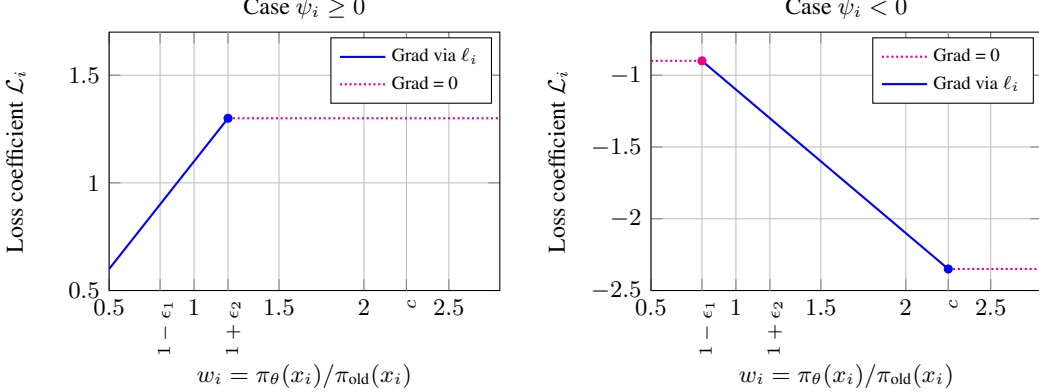

Figure 3: Visualization of the loss coefficient $\mathcal{L}_i$ vs. importance weight $w_i$ based on the specific implementation in Algorithm 2 as RPG-REINFORCE. This version swaps the main branching condition compared to previous versions (branches on $\psi_i > 0$). The plot assumes $\ell_i = -\log \pi_\theta(x_i) = 1$ for visualizing the value of $\mathcal{L}_i$. The line styles indicate the nature of the gradient $\nabla_\theta \mathcal{L}_i$: **Solid blue:** Gradient exists, flowing only via $\ell_i$. The coefficient multiplying $\nabla_\theta \ell_i$ depends on $\text{SG}(w_i)$. **Dotted magenta:** Gradient is zero. This occurs when $\ell_i$ is detached via SG in the loss calculation. Left: Case $\psi_i \geq 0$. Right: Case $\psi_i < 0$.

## 5 EXPERIMENTS

In this section, we empirically evaluate our proposed Regularized Policy Gradient (RPG) framework, including both its fully differentiable (RPG) and REINFORCE-style (RPG-REINFORCE) variants. We compare their performance against established baselines on challenging mathematical reasoning tasks using large language models, including GRPO (Shao et al., 2024) and DAPO (Yu et al., 2025). Our evaluation focuses on task-specific accuracy, training stability, and key training dynamics such as reward, policy entropy, and response length.

**Base Models and Datasets.** We conduct experiments using the Qwen3-4B and Qwen2.5-7B-Instruct models. For training, we utilize the DAPO-Math-17k dataset (Yu et al., 2025) (13.9k English samples). We evaluate on AIME2024, AIME2025, and AMC23, and additionally report results on MinervaMath and OlympiadBench in the Appendix. We compare against baselines including GRPO, DAPO, and REINFORCE++.

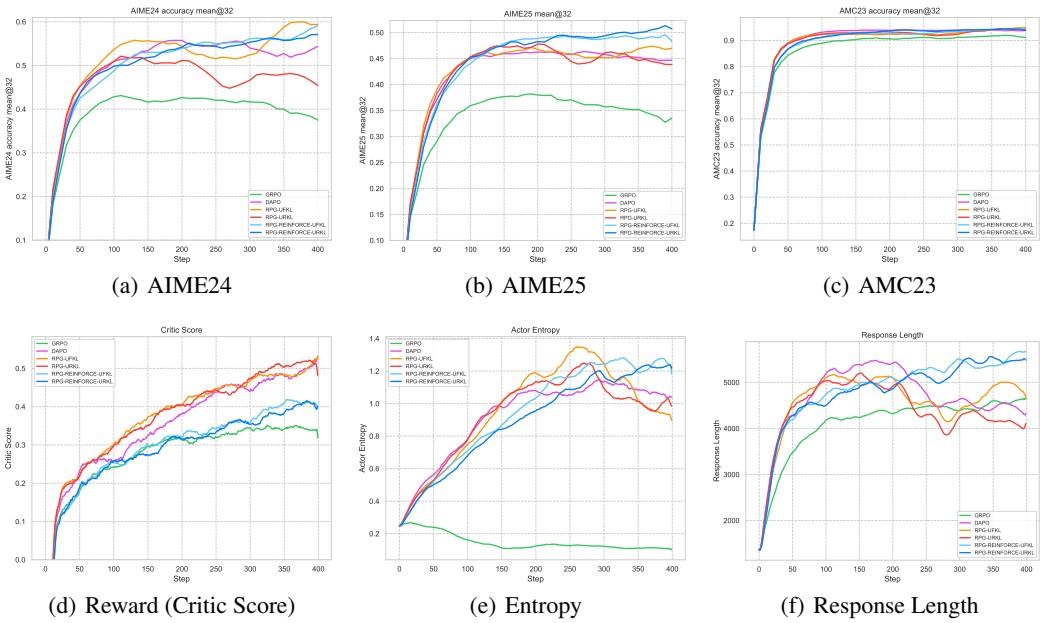

Figure 4: Training dynamics and benchmark performance for fully differentiable Regularized Policy Gradient (RPG) and REINFORCE-Style RPG (RPG-REINFORCE) compared to baselines (GRPO and DAPO) with 8k context length.

Table 3: Combined performance metrics with 8K context length on the AIME24, AIME25, and AMC23 mathematical reasoning benchmarks, showing "Last" and "Best" scores. The "Last" score is from the 400th training step, assuming the training process remained stable to that point. The highest score in each column is **bolded**, and the second highest is underlined. RPG and RPG-REINFORCE methods are highlighted with light cyan and light green backgrounds, respectively.

| Method | AIME24 | | AIME25 | | AMC23 | |
|---|---|---|---|---|---|---|
| | **Last** | **Best** | **Last** | **Best** | **Last** | **Best** |
| GRPO | 0.3750 | 0.4396 | 0.3354 | 0.4063 | 0.9109 | 0.9297 |
| DAPO | 0.5438 | 0.5740 | 0.4469 | 0.4740 | 0.9375 | 0.9430 |
| RPG-UFKL | **0.5938** | **0.6177** | 0.4698 | 0.4865 | **0.9492** | 0.9517 |
| RPG-URKL | 0.4542 | 0.5260 | 0.5261 | 0.4938 | 0.9406 | **0.9539** |
| RPG-REINFORCE-UFKL | 0.5906 | 0.5958 | 0.4833 | 0.5031 | 0.9453 | 0.9469 |
| RPG-REINFORCE-URKL | 0.5708 | 0.5781 | **0.5073** | **0.5208** | 0.9398 | 0.9469 |

**Implementation and Framework.** Experiments are implemented using the `verl` framework (Sheng et al., 2025) with the `vLLM` engine (Kwon et al., 2023) for efficient LLM serving and inference. For practical implementation of our RPG methods, we emphasize that the probabilities (or log-probabilities) from the last iteration's model ($\pi_{\text{old}}$) for the sampled data can be pre-computed and stored. This allows the KL regularization terms to be calculated without needing to keep $\pi_{\text{old}}$ in GPU memory during the training step of the current policy $\pi_\theta$. Consequently, only *one* model ($\pi_\theta$) needs to be actively managed in GPU memory for training, which is faster and more memory-efficient compared to approaches like GRPO that typically require access to at least two models (the current policy and a reference/sampling policy) during optimization.

**Iterative reference updates.** To further stabilize optimization, we adopt an *iterative reference-update* scheme: we periodically set $\pi_{\text{old}} \leftarrow \pi_\theta$ (every $K$ optimizer steps, or when a moving average of token-level KL exceeds a target $\kappa$). This realizes a practical KL trust region while avoiding over-regularization toward the initial checkpoint. Further implementation details and hyperparameters (learning rate, $\beta$, clipping) are provided in Appendix H.

**Stabilization and Advanced RL Techniques.** Our RPG implementations (both fully differentiable and REINFORCE-style) incorporate stabilization techniques like baseline subtraction and PPO-style objective clipping (specifically, Dual-Clip (Ye et al., 2020; Schulman et al., 2017)), crucial for robust off-policy learning. Detailed algorithmic descriptions are provided in Appendix G (see Algorithm 1 for RPG with Dual-Clip and Algorithm 2 for the REINFORCE-style equivalent, along with Figures 3 and 2 for visualization). Varying the clip ratios in REINFORCE-style RPG algorithms, we find that while critic scores and response lengths are similar for $(\epsilon_1, \epsilon_2) = (0.1, 0.1)$ and $(0.2, 0.28)$ (Figure 8), DAPO's higher-and-clip-higher strategy substantially reduces actor entropy, which appears to underlie its performance gains, details can be found in Appendix I.2.1. For PPO-style clipping, we set $(\epsilon_1, \epsilon_2) = (0.2, 0.28)$ for RPG, RPG-REINFORCE and DAPO. For GRPO, we use $(\epsilon_1, \epsilon_2) = (0.2, 0.2)$. Furthermore, to enhance training efficiency and data quality, we adopted techniques introduced by DAPO (Yu et al., 2025), including a dynamic sampling strategy with a group filtering mechanism, which oversamples challenging prompts and filters out those with near-perfect or near-zero accuracy based on initial rollouts and an overlong punishment component in the reward shaping to discourage excessively verbose outputs. In addition, we enable *RPG-Style Clip* (Section 3.3) for the REINFORCE-style estimators, which we found to be the best variant for RL training at larger scales.

**Results and Discussion.** We display the curves in Figure 4 and last and best scores on AIME24 and AIME25 benchmarks in Table 3 for the experiments with 8K context length. The results also demonstrate the superiority of our algorithms over baselines, including GRPO and DAPO. With 8k context length, RPG-REINFORCE achieves 52% accuracy on AIME25, surpassing the official Qwen3-4B-Instruct baseline (47%). Tables 5 and 6 in Appendix I summarize the performance of our RPG algorithms against baselines with 4k and 2k context lengths, reporting both the last and best scores achieved during training on these benchmarks. Figures 5 and 6 in Appendix I complement these results by illustrating the evaluation scores and training dynamics for the fully differentiable RPG variants and baselines when training the Qwen-3-4B model using 4k and 2k context length, respectively. These figures display performance on the AIME24 and AIME25 benchmarks, alongside key training metrics: reward (critic score), policy entropy, and average response length. Across settings, the RPG-REINFORCE variants with RPG-Style Clip have the strongest results. Following DAPO (Yu et al., 2025; Yue et al., 2025), we report "Mean@32" (average accuracy of 32 sampled responses).

Moreover, these algorithms generally exhibit stable training progressions regarding reward (critic score) and policy entropy, as shown in subfigures (c) and (d) in Figures 5 and 6 in the Appendix, compared to some baselines like GRPO, which can show more volatility. This stability likely contributes to their robust benchmark performances (subfigures a-b). The response lengths (subfigure e) for RPG methods also appear well-controlled. These observations align with the strong final scores reported in Tables 5 and 6 for these variants.

## 6 CONCLUSION

We introduced RPG, a framework for deriving and organizing KL-regularized policy gradient algorithms for online, off-policy RL. We provided derivations for policy gradients and surrogate loss functions covering forward/reverse KL, normalized/unnormalized distributions, and both fully differentiable and REINFORCE-style estimators. Beyond derivations, we revisited the classical REINFORCE algorithm and made it viable off-policy through RPG-Style Clip and iterative reference updates. On LLM reasoning, these design choices deliver stable and scalable training with competitive and superior accuracy relative to strong baselines.

### ETHICS STATEMENT

The methods developed in this paper contribute to the broader effort of enhancing the reasoning capabilities of large language models. Improved reasoning in LLMs has the potential to significantly benefit various fields, including scientific discovery, education, and complex problem-solving in engineering and medicine. By providing more stable and efficient training algorithms, our work can facilitate the development of more reliable and capable AI systems.

However, as with any advancement in AI capabilities, it is crucial to consider the ethical implications and ensure responsible development and deployment of these technologies to mitigate potential

misuse. While our framework offers a unified perspective on KL-regularized policy gradient algorithms and demonstrates strong empirical performance, it has certain limitations. RPG-Style Clip introduces a controllable bias: variance trade-off through $(\epsilon_1, \epsilon_2)$, so developing principled schedules for clipping would be valuable. We used LLMs as assistive tools to polish part of this paper. The roles of LLMs in this work are restricted to improving readability and presentation.

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

# Appendix

# A  RELATED WORK

Fine-tuning large language models (LLMs) using human feedback has become a critical step in developing capable and aligned AI systems. Broadly, methods fall into two main categories: those relying on policy optimization using an explicit reward model learned from feedback, and those directly optimizing policies based on preference data.

**RLHF via Policy Optimization.** The classic RLHF involves training a reward model (RM) $r_\phi(x, y)$ to predict human preferences and then using reinforcement learning to optimize the language model policy $\pi_\theta$ to maximize the expected reward from the RM, often regularizing against deviating too far from an initial reference policy $\pi_{\text{ref}}$. This approach was pioneered by Christiano et al. (2017) and gained widespread prominence with its application to LLMs like InstructGPT (Ouyang et al., 2022) and ChatGPT (OpenAI, 2022), which utilized Proximal Policy Optimization (PPO) (Schulman et al., 2017). PPO became a workhorse due to its relative stability, achieved by constraining policy updates via a clipped surrogate objective. The standard PPO setup for RLHF involves the policy $\pi_\theta$, a value function $V_\psi$, the RM $r_\phi$, and the reference policy $\pi_{\text{ref}}$.

**RLHF via Direct Preference Optimization.** An alternative and increasingly popular approach bypasses explicit reward modeling by directly optimizing the policy $\pi_\theta$ based on preference data, typically pairwise comparisons $(y_w, y_l)$ indicating that response $y_w$ is preferred over $y_l$ for a given prompt $x$. Inspired by the Bradley-Terry model (Bradley and Terry, 1952), Direct Preference Optimization (DPO) (Rafailov et al., 2023) derived a simple loss function directly relating preference probabilities to policy likelihoods under $\pi_\theta$ and a reference policy $\pi_{\text{ref}}$. DPO maximizes the relative likelihood of preferred responses using a logistic loss: $\mathcal{L}_{\text{DPO}} \propto -\mathbb{E}[\log \sigma(\beta \Delta \log p)]$, where $\Delta \log p$ is the difference in log-probabilities of $y_w$ and $y_l$ between $\pi_\theta$ and $\pi_{\text{ref}}$. DPO's simplicity and effectiveness led to its wide adoption in models like Llama-3 (Grattafiori et al., 2024), Qwen2 (Yang et al., 2024), and Phi-3 (Abdin et al., 2024). Numerous variants have followed: SLiC-HF (Zhao et al., 2023) uses a pairwise hinge loss for calibration; IPO (Azar et al., 2024) uses an identity link function; SimPO (Meng et al., 2024) offers a simpler objective focusing on the margin; KTO (Ethayarajh et al., 2024) handles binary (good/bad) feedback; DQO (Ji et al., 2024) incorporates direct Q-value modeling; RAFT (Dong et al., 2023), RSO (Liu et al., 2024) and RFT (Yuan et al., 2023) use a rejection sampling perspective. Recognizing that preferences might evolve, iterative methods like Iterative DPO (Xiong et al., 2024), PCO (Xu et al., 2023) and SPIN (Chen et al., 2024) alternate between generation/preference learning and policy updates, often using the current policy's outputs in a self-improvement loop. Game theory offers another lens, with Nash Learning from Human Feedback (NLHF) (Munos et al., 2024) framing RLHF as finding a Nash equilibrium between policies. Self-play ideas appear in SPPO (Wu et al., 2025) and GPO (Zhang et al., 2025), where the policy generates pairs for comparison. Methods like GPM (Zhang et al., 2025) aim to handle more general preference structures efficiently using latent embeddings beyond pairwise comparisons.

**RL for Enhancing LLM Reasoning.** Beyond general alignment with human preferences, RL techniques are increasingly explored to specifically enhance the multi-step reasoning capabilities of LLMs in domains like mathematics, coding, and complex instruction following. In these contexts, RL optimizes the policy to generate sequences (e.g., chain-of-thought, code blocks) that lead to successful outcomes, often using rewards derived from external feedback like unit test results, execution outcomes, or correctness checks by an automated judge or specialized reward model trained on reasoning quality. For instance, the DeepSeekMath model (Shao et al., 2024) employed the GRPO algorithm, a value-free PPO variant, demonstrating significant improvements in mathematical problem-solving benchmarks through RL fine-tuning. DeepSeek-R1 (Guo et al., 2025) represents efforts in applying advanced techniques potentially involving RL for complex tasks, although specific methods might vary. Furthermore, preference-based methods like SPPO and GPO have been applied to reasoning-specialized models such as Kimi-1.5 (Team et al., 2025), and the resulting improvements observed on benchmarks involving coding and math suggest that preference-based RLHF can also contribute to refining reasoning abilities, potentially by optimizing implicit properties related to logical consistency and correctness within the preference data. The need for a value function (critic model) used in PPO incurs significant computational costs, and standard PPO can face stability challenges with sparse rewards common in LLM tasks. Addressing these issues has driven recent work. Several methods aim to improve efficiency by removing the value network: RLOO (Kool et al., 2019; Ahmadian et al., 2024) shows that drawing multiple samples per input allows for a baseline based on the average reward. ReMax (Li et al., 2024) adapts REINFORCE (Williams, 1992)

using Monte Carlo returns and normalization; GRPO (Shao et al., 2024) uses a group-average reward baseline and adds a $k_3$-based KL penalty to the objective; and VinePPO (Kazemnejad et al., 2024) uses MC sampling from intermediate steps. Other approaches focus on stability and alternative baselines, such as RLOO (Ahmadian et al., 2024), which uses leave-one-out statistics within a group, and REINFORCE++ (Hu, 2025), which enhances REINFORCE with token-level KL penalties (using the $k_2$ estimator) and normalization. Dr. GRPO (Liu et al., 2025) identifies and corrects a bias found in GRPO's advantage estimators, DAPO (Yu et al., 2025) introduces strategies like Clip-Higher, reward over-sampling, and a token-level loss to handle long sequences and entropy collapse, while VAPO (Yuan et al., 2025) builds upon it with length-adaptive advantage estimation. Group Policy Gradient (GPG) (Chu et al., 2025) revisits the original REINFORCE objective, using group-normalized rewards and a debiased gradient estimator. Recently, GSPO (Zheng et al., 2025) was proposed with sequence-level rewards and used in the Qwen3 model series (Team, 2025).

Our contribution is to make the off-policy weighting and estimator equivalences explicit across normalized/unnormalized variants, to identify a bias introduced when these weights are omitted (as in the GRPO KL term), and to provide corrected surrogates that are gradient-equivalent to the intended objectives. The design-space view makes transparent how several recent algorithms arise as special cases.

## B CONNECTION BETWEEN REGULARIZED POLICY GRADIENT AND NATURAL POLICY GRADIENT

In this section, we draw the connection between RPG and Natural Policy Gradient (NPG) (Kakade, 2001; Schulman et al., 2015). Note that NPG moves along the steepest-ascent direction defined by the Riemannian geometry induced by the Fisher information matrix, rather than by the Euclidean geometry of the raw parameters. More specifically, we demonstrate that the Natural Policy Gradient (NPG) update can be recovered as a special instance of the RPG update by applying a linear approximation to the expected return and a quadratic approximation to the KL regularization term.

In detail, consider the RPG objective at iteration $k$:

$$\mathcal{J}_{\mathrm{RPG}}(\theta) = J(\theta) - \beta \ \mathrm{KL}(\pi_{\theta_k} \| \pi_\theta), \tag{B.1}$$

where $J(\theta)$ is the expected return. To study the local behavior of the update, we first apply the first-order Taylor expansion to the return $J(\theta)$ around the current policy parameter $\theta_k$:

$$J(\theta) \approx J(\theta_k) + \nabla_\theta J(\theta_k)^\top \Delta\theta, \tag{B.2}$$

where $\Delta\theta = \theta - \theta_k$ denotes a small change.

Then we apply the second-order Taylor expansion of the KL divergence term at $\theta_k$ as follows:

$$\mathrm{KL}(\pi_{\theta_k} \| \pi_\theta) \approx \mathrm{KL}(\pi_{\theta_k} \| \pi_{\theta_k}) + \nabla_\theta \ \mathrm{KL}(\pi_{\theta_k} \| \pi_\theta)\big|_{\theta=\theta_k}^\top \Delta\theta$$
$$+ \frac{1}{2}\Delta\theta^\top \nabla_\theta^2 \ \mathrm{KL}(\pi_{\theta_k} \| \pi_\theta)\big|_{\theta=\theta_k} \Delta\theta \tag{B.3}$$
$$= 0 + 0 + \frac{1}{2}\Delta\theta^\top F(\theta_k)\Delta\theta, \tag{B.4}$$

where $\Delta\theta = \theta - \theta_k$, and $F(\theta_k)$ is the Fisher information matrix:

$$F(\theta_k) = \mathbb{E}_{x \sim \pi_{\theta_k}} \left[ \nabla_\theta \log \pi_\theta(x)\big|_{\theta=\theta_k} \nabla_\theta \log \pi_\theta(x)\big|_{\theta=\theta_k}^\top \right].$$

Note that the Fisher information matrix describes the local geometry of the parameter space of the policy family. It gives a good metric inside a small neigbourhood around $\theta_k$.

Now insert (B.2) and (B.4) back into the RPG objective (B.1), we obtain the following quadratic surrogate $\widetilde{\mathcal{J}}_{\mathrm{RPG}}(\Delta\theta)$ around $\theta_k$:

$$\widetilde{\mathcal{J}}_{\mathrm{RPG}}(\Delta\theta) = J(\theta_k) + \nabla_\theta J(\theta_k)^\top \Delta\theta - \frac{\beta}{2}\Delta\theta^\top F(\theta_k)\Delta\theta. \tag{B.5}$$

We now look for the best local step $\Delta\theta^*$. Take the gradient of (B.5) with respect to $\Delta\theta$ and set it equal to zero, we obtain:

$$\nabla_{\Delta\theta}\widetilde{\mathcal{J}}_{\text{RPG}} = \nabla_\theta J(\theta_k) - \beta F(\theta_k)\Delta\theta = 0.$$

Solving this linear system for $\Delta\theta$ gives

$$\Delta\theta^* = \frac{1}{\beta}F(\theta_k)^{-1}\nabla_\theta J(\theta_k).$$

The step $\Delta\theta^*$ matches the natural policy gradient (Kakade, 2001) update direction up to the factor $1/\beta$. This suggests that the policy gradient update of RPG in (B.1) can be approximated by NPG as follows

$$\theta_{k+1} \leftarrow \theta_k + \frac{1}{\beta}F(\theta_k)^{-1}\nabla_\theta J(\theta_k).$$

In other words, the maximizer of the local KL regularized RPG approximation follows the same direction as a natural policy gradient update.

## C  REINFORCE AND PROXIMAL POLICY OPTIMIZATION (PPO)

### C.1  REINFORCE

REINFORCE performs Monte Carlo (MC) updates after sampling a complete trajectory, using the sampled return $G_t$ as an unbiased estimate of the state-action value function $Q^{\pi_\theta}(s_t, a_t)$. However, these MC estimates often exhibit high variance, leading to slow and unstable learning.

To reduce variance, a state-dependent baseline $b(s_t)$ (commonly an estimate of the state value function, $V^{\pi_\theta}(s_t)$) is subtracted from the return-to-go:

$$\nabla_\theta J(\theta) = \mathbb{E}_{\tau\sim\pi_\theta}\left[\sum_{t=0}^{T}(G_t - b(s_t))\nabla_\theta \log \pi_\theta(a_t|s_t)\right] = \mathbb{E}_{\tau\sim\pi_\theta}\left[\sum_{t=0}^{T}\widehat{A}_t\nabla_\theta \log \pi_\theta(a_t|s_t)\right]. \tag{C.1}$$

Here, $\widehat{A}_t = G_t - b(s_t)$ is an estimate of the advantage function $A^{\pi_\theta}(s_t, a_t) = Q^{\pi_\theta}(s_t, a_t) - V^{\pi_\theta}(s_t)$. Subtracting a baseline that only depends on the state $s_t$ does not bias the gradient estimate, since $\mathbb{E}_{a_t\sim\pi_\theta(\cdot|s_t)}[b(s_t)\nabla_\theta \log \pi_\theta(a_t|s_t)] = b(s_t)\nabla_\theta \sum_{a_t}\pi_\theta(a_t|s_t) = b(s_t)\nabla_\theta 1 = 0$. REINFORCE with baseline is typically implemented by minimizing the loss:

$$\mathcal{L}_{\text{REINFORCE}}(\theta) = -\mathbb{E}_{\tau\sim\pi_\theta}\left[\sum_{t=0}^{T}\text{SG}(\widehat{A}_t)\log \pi_\theta(a_t|s_t)\right], \tag{C.2}$$

using the stop-gradient operator $\text{SG}(\cdot)$ to prevent gradients from flowing into the advantage estimate $\widehat{A}_t$. As REINFORCE uses samples collected under the current policy $\pi_\theta$ for gradient estimation, it is an on-policy algorithm.

### C.2  PROXIMAL POLICY OPTIMIZATION (PPO)

On-policy methods like REINFORCE can be sample-inefficient, requiring new trajectories for each gradient update. Proximal Policy Optimization (PPO) (Schulman et al., 2017) improves stability and sample efficiency by enabling multiple updates using the same batch of data collected under a slightly older policy $\pi_{\theta_{\text{old}}}$. This makes PPO effectively off-policy. PPO achieves this by optimizing a surrogate objective function that discourages large deviations between the current policy $\pi_\theta$ and the old policy $\pi_{\theta_{\text{old}}}$. The most widely used variant, PPO-Clip, employs a clipped objective:

$$J^{\text{PPO-Clip}}(\theta) = \mathbb{E}_t\left[\min\left(w_t(\theta)\widehat{A}_t, \text{clip}(w_t(\theta), 1-\epsilon, 1+\epsilon)\widehat{A}_t\right)\right], \tag{C.3}$$

where the expectation $\mathbb{E}_t$ is taken over timesteps in the collected batch sampled from $\pi_{\text{old}}$. Here, $w_t(\theta) = \frac{\pi_\theta(a_t|s_t)}{\pi_{\text{old}}(a_t|s_t)}$ is the importance sampling ratio. $\widehat{A}_t$ is an advantage estimate, typically computed

using Generalized Advantage Estimation (GAE) (Schulman et al., 2016), which leverages observed rewards and a learned state-value function $V(s)$ to reduce variance.

Notably, in many practical implementations, especially in Reinforcement Learning from Human Feedback (RLHF) for large language models (Ouyang et al., 2022), a KL divergence penalty against a reference policy $\pi_{\text{ref}}$ (e.g., the initial supervised model) is often incorporated implicitly by modifying the reward signal before calculating the advantage. For example, the reward used for GAE calculation might become $r'_t = r_t - \beta \log(\pi_\theta(a_t|s_t)/\pi_{\text{ref}}(a_t|s_t))$. When this $r'_t$ is used within GAE to compute $\widehat{A}_t$, the KL penalty term is effectively folded into the advantage estimate that multiplies the importance weight $w_t(\theta)$ in the objective function. This approach contrasts with adding the KL penalty as a separate term to the final objective, as seen in GRPO (Section 2.2) or the formal derivations in Section 3.

The hyperparameter $\epsilon$ (e.g., 0.2) defines the clipping range $[1 - \epsilon, 1 + \epsilon]$ for the importance ratio $w_t(\theta)$. This clipping limits the influence of potentially noisy importance weights when the policy changes significantly, preventing destructive updates and further stabilizing the off-policy training. PPO optimizes the policy $\pi_\theta$ by maximizing $J^{\text{PPO-Clip}}(\theta)$.

## D    EQUIVALENCE OF $k_3$ ESTIMATOR AND UNNORMALIZED KL DIVERGENCE

As mentioned in Section 3.2, the $k_3$ estimator for KL divergence (Schulman, 2020) is equivalent to the unnormalized KL (UKL) divergence. The $k_3$ function is defined as $k_3(y) = y - 1 - \log y$.

**Forward KL-$k_3$ and** $\text{UKL}(\pi_{\text{old}}\|\pi_\theta)$: The forward KL-$k_3$ divergence is $\text{KL}_{k_3}(\pi_{\text{old}}\|\pi_\theta) := \mathbb{E}_{x\sim\pi_{\text{old}}}[k_3(\pi_\theta(x)/\pi_{\text{old}}(x))]$.

$$
\begin{aligned}
\mathbb{E}_{x\sim\pi_{\text{old}}}\left[k_3\left(\frac{\pi_\theta(x)}{\pi_{\text{old}}(x)}\right)\right] &= \mathbb{E}_{x\sim\pi_{\text{old}}}\left[\frac{\pi_\theta(x)}{\pi_{\text{old}}(x)} - 1 - \log\frac{\pi_\theta(x)}{\pi_{\text{old}}(x)}\right] \\
&= \int_x \pi_{\text{old}}(x)\left(\frac{\pi_\theta(x)}{\pi_{\text{old}}(x)} - 1\right)dx - \int_x \pi_{\text{old}}(x)\log\frac{\pi_\theta(x)}{\pi_{\text{old}}(x)}dx \\
&= \int_x (\pi_\theta(x) - \pi_{\text{old}}(x))dx + \int_x \pi_{\text{old}}(x)\log\frac{\pi_{\text{old}}(x)}{\pi_\theta(x)}dx \\
&= \text{UKL}(\pi_{\text{old}}\|\pi_\theta).
\end{aligned}
$$

**Reverse KL-$k_3$ and** $\text{UKL}(\pi_\theta\|\pi_{\text{old}})$: The reverse KL-$k_3$ divergence is $\text{KL}_{k_3}(\pi_\theta\|\pi_{\text{old}}) := \mathbb{E}_{x\sim\pi_\theta}[k_3(\pi_{\text{old}}(x)/\pi_\theta(x))]$.

$$
\begin{aligned}
\mathbb{E}_{x\sim\pi_\theta}\left[k_3\left(\frac{\pi_{\text{old}}(x)}{\pi_\theta(x)}\right)\right] &= \mathbb{E}_{x\sim\pi_\theta}\left[\frac{\pi_{\text{old}}(x)}{\pi_\theta(x)} - 1 - \log\frac{\pi_{\text{old}}(x)}{\pi_\theta(x)}\right] \\
&= \int_x \pi_\theta(x)\left(\frac{\pi_{\text{old}}(x)}{\pi_\theta(x)} - 1\right)dx - \int_x \pi_\theta(x)\log\frac{\pi_{\text{old}}(x)}{\pi_\theta(x)}dx \\
&= \int_x (\pi_{\text{old}}(x) - \pi_\theta(x))dx + \int_x \pi_\theta(x)\log\frac{\pi_\theta(x)}{\pi_{\text{old}}(x)}dx \\
&= \text{UKL}(\pi_\theta\|\pi_{\text{old}}).
\end{aligned}
$$

## E    NORMALIZED KL REGULARIZATION

For completeness, we collect here the normalized KL formulations that were previously in the main text. Their proofs remain in Appendix K.

### E.1    FORWARD KL REGULARIZATION

Consider the objective function with forward KL regularization:

$$J_{\text{FKL}}(\theta) = \mathbb{E}_{x\sim\pi_\theta}[R(x)] - \beta\,\text{KL}(\pi_{\text{old}} \| \pi_\theta). \tag{E.1}$$

**Proposition E.1** (Policy Gradient and Differentiable Loss for Forward KL). The gradient of $J_{\text{FKL}}(\theta)$ with respect to $\theta$ is:

$$\nabla_\theta J_{\text{FKL}}(\theta) = \mathbb{E}_{x\sim\pi_{\text{old}}}\left[\left(w(x)R(x) + \beta\right)\nabla_\theta \log\pi_\theta(x)\right],$$

Table 4: Summary of fully differentiable surrogate losses for normalized KL-regularized objectives (counterparts to Table 1). Here $x \sim \pi_{\mathrm{old}}$, $w(x) = \pi_\theta(x)/\pi_{\mathrm{old}}(x)$.

| Regularization (Normalized KL) | Surrogate loss (sampling $x \sim \pi_{\mathrm{old}}$) |
|:---:|:---:|
| Forward KL | $\mathbb{E}[-w(x)R(x) - \beta \log \pi_\theta(x)]$ |
| Reverse KL | $\mathbb{E}[w(x)\left(-R(x) + \beta \log w(x)\right)]$ |

where $w(x) = \pi_\theta(x)/\pi_{\mathrm{old}}(x)$. A corresponding surrogate loss is:

$$\mathcal{L}_{\mathrm{FKL}}(\theta) = \mathbb{E}_{x\sim\pi_{\mathrm{old}}}\left[ - w(x)R(x) - \beta \log \pi_\theta(x)\right],$$

which satisfies $\nabla_\theta \mathcal{L}_{\mathrm{FKL}}(\theta) = -\nabla_\theta J_{\mathrm{FKL}}(\theta)$.

**Remark E.2** (Connection to Maximum Likelihood Estimation). If $R(x) = 0$, maximizing $J_{\mathrm{FKL}}(\theta)$ reduces to minimizing $\beta \mathrm{KL}(\pi_{\mathrm{old}} \parallel \pi_\theta)$, i.e., MLE on samples from $\pi_{\mathrm{old}}$.

### E.2 REVERSE KL REGULARIZATION

Consider the reverse KL objective:

$$J_{\mathrm{RKL}}(\theta) = \mathbb{E}_{x\sim\pi_\theta}[R(x)] - \beta \mathrm{KL}(\pi_\theta \parallel \pi_{\mathrm{old}}). \tag{E.2}$$

**Proposition E.3** (Policy Gradient and Differentiable Loss for Reverse KL). The gradient of $J_{\mathrm{RKL}}(\theta)$ is:

$$\nabla_\theta J_{\mathrm{RKL}}(\theta) = \mathbb{E}_{x\sim\pi_{\mathrm{old}}}\left[w(x)\Big(R(x) - \beta(\log w(x) + 1)\Big)\nabla_\theta \log \pi_\theta(x)\right].$$

A corresponding surrogate loss is:

$$\mathcal{L}_{\mathrm{RKL}}(\theta) = \mathbb{E}_{x\sim\pi_{\mathrm{old}}}\left[w(x)\big(-R(x) + \beta \log w(x)\big)\right],$$

with $\nabla_\theta \mathcal{L}_{\mathrm{RKL}}(\theta) = -\nabla_\theta J_{\mathrm{RKL}}(\theta)$.

**REINFORCE-style RPG with normalized KL regularizations.** REINFORCE-style losses for FKL/RKL appear in Appendix F (Table analogues to Table 2).

## F REINFORCE-STYLE REGULARIZED POLICY GRADIENTS WITH VARIOUS KL REGULARIZATION FORMS

### F.1 RATIONALE FOR REINFORCE-STYLE LOSS FORMULATION

As noted in Section 4 of the main text, the derived off-policy policy gradients (Theorems E.1 through 3.6) share a structural similarity with the REINFORCE estimator:

$$\nabla_\theta J(\theta) = \mathbb{E}_{x\sim\pi_{\mathrm{sampling}}}\left[\mathrm{Weight}(x,\theta)\nabla_\theta \log \pi_\theta(x)\right].$$

This structure suggests an alternative way to implement the gradient update, analogous to the REINFORCE-style approach used in the on-policy setting. Specifically, one could define a surrogate loss of the form:

$$\mathcal{L}_{\mathrm{REINFORCE\text{-}style}}(\theta) = -\mathbb{E}_{x\sim\pi_{\mathrm{sampling}}}\left[\mathrm{SG}\left(\mathrm{Weight}(x,\theta)\right) \log \pi_\theta(x)\right]. \tag{F.1}$$

The rationale is that applying automatic differentiation to this loss should yield:

$$\nabla_\theta \mathcal{L}_{\mathrm{REINFORCE\text{-}style}}(\theta) \overset{\mathrm{Autodiff}}{=} -\mathbb{E}_{x\sim\pi_{\mathrm{sampling}}}\left[\mathrm{SG}\left(\mathrm{Weight}(x,\theta)\right) \nabla_\theta \log \pi_\theta(x)\right].$$

When this gradient is used for optimization, the stop-gradient SG is conceptually removed, resulting in an update aligned with $-\nabla_\theta J(\theta)$. This relies on SG preventing gradients from flowing through the $\theta$-dependence within $\mathrm{Weight}(x,\theta)$ (specifically, the dependence via the importance weight $w(x)$). The following subsections detail these REINFORCE-style loss formulations for each KL regularization type.

## F.2 REINFORCE-STYLE RPG WITH FORWARD KL REGULARIZATION

We can convert Forward KL regularization of RPG to REINFORCE-style using the stop-gradient operator:

**Proposition F.1** (REINFORCE-Style Loss for Forward KL)**.** For the forward KL regularized objective function in Eq. (E.1), the corresponding REINFORCE-style surrogate loss function for gradient descent optimization via automatic differentiation is:

$$\mathcal{L}_{\text{FKL}}^{\text{REINFORCE-style}}(\theta) = -\mathbb{E}_{x \sim \pi_{\text{old}}} \left[ \text{SG} \left( w(x)R(x) + \beta \right) \log \pi_\theta(x) \right],$$

where $w(x) = \pi_\theta(x)/\pi_{\text{old}}(x)$. This loss aims to produce the gradient $-\nabla_\theta J_{\text{FKL}}(\theta)$ via automatic differentiation.

**Remark F.2.** This REINFORCE-style loss requires SG to prevent backpropagation through $w(x)$ in the weight term. Baselines can be added to $R(x)$ inside SG for variance reduction (see Appendix G). In practice we further apply *RPG-Style Clip* (Section 3.3) by replacing $w$ with $\bar{w}$ and, when present, $\log w$ with $\log \bar{w}$ inside $\text{SG}(\cdot)$.

## F.3 REINFORCE-STYLE RPG WITH UNNORMALIZED FORWARD KL REGULARIZATION

Similarly, we can also transform the Unnormalized Forward KL Regularization of RPG into REINFORCE-style as follows:

**Proposition F.3** (REINFORCE-Style Loss for Unnormalized Forward KL)**.** For the objective $J_{\text{UFKL}}(\theta) = \mathbb{E}_{\pi_\theta}[R(x)] - \beta \, \text{UKL}(\pi_{\text{old}} \| \pi_\theta)$, whose gradient (sampling from $\widetilde{\pi}_{\text{old}}$) is $\nabla_\theta J_{\text{UFKL}}(\theta) = \mathbb{E}_{x \sim \widetilde{\pi}_{\text{old}}}[Z_{\text{old}}(w(x)R(x) - \beta(w(x) - 1))\nabla_\theta \log \pi_\theta(x)]$ (Proposition 3.2), a corresponding REINFORCE-style surrogate loss is:

$$\mathcal{L}_{\text{UFKL}}^{\text{REINFORCE-style}}(\theta) = -\mathbb{E}_{x \sim \widetilde{\pi}_{\text{old}}} \left[ \text{SG} \left( Z_{\text{old}} \left( w(x)R(x) - \beta(w(x) - 1) \right) \right) \log \pi_\theta(x) \right],$$

where $\widetilde{\pi}_{\text{old}} = \pi_{\text{old}}/Z_{\text{old}}$ and $w(x) = \pi_\theta(x)/\pi_{\text{old}}(x)$ (using unnormalized $\pi_{\text{old}}$). This loss aims to produce the gradient $-\nabla_\theta J_{\text{UFKL}}(\theta)$ via automatic differentiation.

## F.4 REINFORCE-STYLE RPG WITH REVERSE KL REGULARIZATION

**Proposition F.4** (REINFORCE-Style Loss for Reverse KL)**.** For the objective $J_{\text{RKL}}(\theta) = \mathbb{E}_{\pi_\theta}[R(x)] - \beta \, \text{KL}(\pi_\theta \| \pi_{\text{old}})$, whose gradient is $\nabla_\theta J_{\text{RKL}}(\theta) = \mathbb{E}_{x \sim \pi_{\text{old}}}[w(x)(R(x) - \beta(\log w(x) + 1))\nabla_\theta \log \pi_\theta(x)]$ (Proposition E.3), a corresponding REINFORCE-style surrogate loss is:

$$\mathcal{L}_{\text{RKL}}^{\text{REINFORCE-style}}(\theta) = -\mathbb{E}_{x \sim \pi_{\text{old}}} \left[ \text{SG} \left( w(x) \left( R(x) - \beta \log w(x) - \beta \right) \right) \log \pi_\theta(x) \right], \qquad \text{(F.2)}$$

where $w(x) = \pi_\theta(x)/\pi_{\text{old}}(x)$. This loss aims to produce the gradient $-\nabla_\theta J_{\text{RKL}}(\theta)$ via automatic differentiation.

## F.5 REINFORCE-STYLE RPG WITH UNNORMALIZED REVERSE KL REGULARIZATION

**Proposition F.5** (REINFORCE-Style Loss for Unnormalized Reverse KL)**.** For the objective $J_{\text{URKL}}(\theta) = \mathbb{E}_{\pi_\theta}[R(x)] - \beta \, \text{UKL}(\pi_\theta \| \pi_{\text{old}})$, whose gradient (sampling from $\widetilde{\pi}_{\text{old}}$) is $\nabla_\theta J_{\text{URKL}}(\theta) = \mathbb{E}_{x \sim \widetilde{\pi}_{\text{old}}}[Z_{\text{old}}w(x)(R(x) - \beta \log w(x))\nabla_\theta \log \pi_\theta(x)]$ (Proposition 3.6), a corresponding REINFORCE-style surrogate loss is:

$$\mathcal{L}_{\text{URKL}}^{\text{REINFORCE-style}}(\theta) = -\mathbb{E}_{x \sim \widetilde{\pi}_{\text{old}}} \left[ \text{SG} \left( Z_{\text{old}}w(x) \left( R(x) - \beta \log w(x) \right) \right) \log \pi_\theta(x) \right],$$

where $\widetilde{\pi}_{\text{old}} = \pi_{\text{old}}/Z_{\text{old}}$ and $w(x) = \pi_\theta(x)/\pi_{\text{old}}(x)$ (using unnormalized $\pi_{\text{old}}$). This loss aims to produce the gradient $-\nabla_\theta J_{\text{URKL}}(\theta)$ via automatic differentiation.

## G MORE ON ALGORITHMIC DETAILS

### G.1 STABILIZATION TECHNIQUES FOR REGULARIZED POLICY GRADIENTS

Practical implementations of off-policy policy gradient methods often require stabilization techniques to manage variance or prevent destructively large policy updates. Common techniques include:

- **Dual-Clip Objective:** This method adapts the clipping mechanism from PPO, with a modification for negative advantages proposed by Ye et al. (2020), to stabilize updates (Schulman et al., 2017). The Dual Clip objective aims to maximize $J^{\text{DualClip}} = \mathbb{E}_{x \sim \pi_{\text{old}}}[L^{\text{DualClip}}(x, \theta)]$, where $\widehat{A}(x)$ is an estimate of the advantage analogue (e.g., $R(x) - b$ or the full term derived from the regularized gradient), $w(x) = \pi_\theta(x)/\pi_{\text{old}}(x)$ is the importance ratio, and $L^{\text{DualClip}}(x, \theta)$ is defined as:

  - If $\widehat{A}(x) \geq 0$: $L^{\text{DualClip}}(x, \theta) = \min(w(x)\widehat{A}(x), \text{ clip}(w(x), 1 - \epsilon_1, 1 + \epsilon_2)\widehat{A}(x))$.

  - If $\widehat{A}(x) < 0$: $L^{\text{DualClip}}(x, \theta) = \max(\min(w(x)\widehat{A}(x), \text{ clip}(w(x), 1 - \epsilon_1, 1 + \epsilon_2)\widehat{A}(x)), c\widehat{A}(x))$.

  where $\epsilon_1, \epsilon_2 > 0$ are clipping parameters and $c > 1$ provides a lower bound for negative advantages. To use this with gradient descent (which minimizes a loss $\mathcal{L}$), we minimize the negative of the Dual Clip objective term. Using $-\min(a, b) = \max(-a, -b)$ and $-\max(a, b) = \min(-a, -b)$, the corresponding loss term for a single sample $x$ is:

  - If $\widehat{A}(x) \geq 0$: $\mathcal{L}^{\text{DualClip}}(x, \theta) = \max\left(-w(x)\widehat{A}(x), -\text{clip}(w(x), 1 - \epsilon_1, 1 + \epsilon_2)\widehat{A}(x)\right)$.

  - If $\widehat{A}(x) < 0$: Let $L_{\text{clip}} = \max\left(-w(x)\widehat{A}(x), -\text{clip}(w(x), 1 - \epsilon_1, 1 + \epsilon_2)\widehat{A}(x)\right)$. Then,

  $$\mathcal{L}^{\text{DualClip}}(x, \theta) = \min\left(L_{\text{clip}}, -c\widehat{A}(x)\right).$$

  Here, $\widehat{A}(x)$ should represent the advantage or an analogous term derived from the gradient of the original (non-negated) regularized objective (e.g., Proposition E.3). The overall loss is $\mathcal{L}(\theta) = \mathbb{E}_{x \sim \pi_{\text{old}}}[\mathcal{L}^{\text{DualClip}}(x, \theta)]$. This loss function is differentiable with respect to $\theta$ (which appears in $w(x)$ and potentially $\widehat{A}(x)$ if it includes terms like $\log w(x)$).

  This loss formulation ensures that updates are conservative. For positive advantages, it acts like standard PPO-Clip. For negative advantages, it prevents the objective from becoming arbitrarily large (loss becoming arbitrarily small) by introducing the lower bound $c\widehat{A}(x)$ on the objective (upper bound $-c\widehat{A}(x)$ on the loss).

- **Baseline Subtraction:** Used to define the advantage $\widehat{A}(x) = R(x) - b(x)$, reducing the variance of the gradient estimates. The baseline $b(x)$ should ideally not depend strongly on $\theta$. A common choice is a value function estimate $V(x)$ or simply the batch average reward $b = \frac{1}{N}\sum R(x_i)$. The definition of $\widehat{A}(x)$ might also incorporate regularization terms depending on the base objective chosen (see RKL example below).

For instance, applying Dual Clip to stabilize the reverse KL objective (Proposition E.3). The gradient involves the term $w(x) \underbrace{\left((R(x) - b) - \beta(\log w(x) + 1)\right)}_{\text{Analogue to } \widehat{A}_{\text{RKL}}(x, w; b)} \nabla \log \pi_\theta$. Using this $\widehat{A}_{\text{RKL}}$ in the Dual Clip loss structure $\mathcal{L}_{\text{RKL}}^{\text{DualClip}}(\theta) = \mathbb{E}_{x \sim \pi_{\text{old}}}[\mathcal{L}_{\text{RKL}}^{\text{DualClip}}(x, \theta)]$ where:

- If $\widehat{A}_{\text{RKL}}(x, w; b) \geq 0$:

$$\mathcal{L}_{\text{RKL}}^{\text{DualClip}}(x, \theta) = \max\left(-w(x)\widehat{A}_{\text{RKL}}, -\text{clip}(w(x), 1 - \epsilon_1, 1 + \epsilon_2)\widehat{A}_{\text{RKL}}\right).$$

- If $\widehat{A}_{\text{RKL}}(x, w; b) < 0$: Let $L_{\text{clip}} = \max\left(-w(x)\widehat{A}_{\text{RKL}}, -\text{clip}(w(x), 1 - \epsilon_1, 1 + \epsilon_2)\widehat{A}_{\text{RKL}}\right)$.

$$\mathcal{L}_{\text{RKL}}^{\text{DualClip}}(x, \theta) = \min\left(L_{\text{clip}}, -c\widehat{A}_{\text{RKL}}\right),$$

where $\widehat{A}_{\text{RKL}}(x, w; b) = (R(x) - b) - \beta(\log w(x) + 1)$. Simpler approximations might use $\widehat{A}(x) = R(x) - b$.

Using PPO-style clipping alters the optimization objective compared to the original KL-regularized objectives, trading strict adherence for enhanced stability. The choice of base objective structure, definition of $\widehat{A}$, and stabilization techniques depends on the specific application.

---

**Algorithm 1** RPG with Dual-Clip Stabilization

---

**Require:** Reference policy $\pi_{\text{old}}$, Reward function $R(x)$, Initial policy parameters $\theta_0$
**Require:** Base objective structure $J_{\text{chosen}}$ (implies regularization type), Regularization strength $\beta \geq 0$
**Require:** Learning rate $\alpha > 0$, Batch size $N > 0$, Number of epochs $K \geq 1$ per iteration
**Require:** Dual Clip parameters: $\epsilon_1 > 0, \epsilon_2 > 0, c > 1$
**Require:** Baseline method (e.g., batch/group average, value function $V_\phi$)
 1: Initialize policy parameters $\theta \leftarrow \theta_0$
 2: Initialize value function parameters $\phi$ (if baseline uses $V_\phi$)
 3: **for** each training iteration **do**
 4:     Sample batch $\mathcal{D} = \{x_i\}_{i=1}^N \sim \pi_{\text{old}}$                 $\triangleright$ Collect data using old policy
 5:     Compute $R_i$ for $i = 1..N$
 6:     Compute baselines $b_i$ for $i = 1..N$ (e.g., $b_i = \frac{1}{N}\sum_j R_j$ or $b_i = V_\phi(x_i)$)
 7:     **for** $k = 1$ to $K$ **do**              $\triangleright$ Multiple optimization epochs on the same batch
 8:         Initialize batch loss $\mathcal{L}_{\text{batch}} = 0$
 9:         **for** $i = 1$ to $N$ **do**
10:             $w_i = \frac{\pi_\theta(x_i)}{\pi_{\text{old}}(x_i)}, \log w_i = \log \pi_\theta(x_i) - \log \pi_{\text{old}}(x_i)$     $\triangleright$ Compute importance weight
11:             Define Advantage analogue $\widehat{A}_i$ based on $J_{\text{chosen}}, R_i, b_i, w_i, \beta$.
12:                 $\triangleright$ Ex: For RKL, $\widehat{A}_i = (R_i - b_i) - \beta(\log w_i + 1)$. Note: $\widehat{A}_i$ depends on current $\theta$ via $w_i$
13:             **if** Dual Clip enabled **then**
14:                 loss_term1$_i = -w_i \times \widehat{A}_i$     $\triangleright$ Negative of unclipped term, gradient flows through $w_i$
15:                 $w_{i,\text{clipped}} = \text{clip}(w_i, 1 - \epsilon_1, 1 + \epsilon_2)$
16:                 loss_term2$_i = -w_{i,\text{clipped}} \times \widehat{A}_i$         $\triangleright$ Negative of clipped term
17:                 $L_{\text{clip}}(i) = \max(\text{loss\_term1}_i, \text{loss\_term2}_i)$
18:                 **if** $\widehat{A}_i \geq 0$ **then**
19:                     $\mathcal{L}_{\text{term}}(i) = L_{\text{clip}}(i)$
20:                 **else**                             $\triangleright \widehat{A}_i < 0$
21:                     loss_lower_bound$_i = -c \times \widehat{A}_i$     $\triangleright$ Lower bound term
22:                     $\mathcal{L}_{\text{term}}(i) = \min(L_{\text{clip}}(i), \text{loss\_lower\_bound}_i)$
23:                 **end if**
24:             **else**
25:                 $\triangleright$ Define base loss term (unclipped) based on chosen objective's negative gradient structure
26:                       $\triangleright$ Ex: For RKL loss (no clip): $\mathcal{L}_{\text{term}}(i) = w_i(-(R_i - b_i) + \beta \log w_i)$
27:                 $\mathcal{L}_{\text{term}}(i) = -w_i \times \widehat{A}_i$
28:             **end if**
29:             $\mathcal{L}_{\text{batch}} = \mathcal{L}_{\text{batch}} + \mathcal{L}_{\text{term}}(i)$
30:         **end for**
31:         $\widehat{\mathcal{L}}(\theta) = \frac{1}{N}\mathcal{L}_{\text{batch}}$         $\triangleright$ Compute final batch loss for minimization
32:         $g \leftarrow \nabla_\theta \widehat{\mathcal{L}}(\theta)$         $\triangleright$ Compute gradient (flows through $w_i$ and $\widehat{A}_i$)
33:         $\theta \leftarrow \text{OptimizerUpdate}(\theta, g, \alpha)$         $\triangleright$ Update policy parameters
34:         **if** using a learned baseline $V_\phi$ **then**
35:             Update value function parameters $\phi$ (e.g., by minimizing $\mathbb{E}[(V_\phi(x_i) - R_i)^2]$ over the batch)
36:         **end if**
37:     **end for**
38: **end for**
39: **return** Optimized policy parameters $\theta$

---

## G.2 STABILIZATION TECHNIQUES FOR REINFORCE-STYLE REGULARIZED POLICY GRADIENTS

While the REINFORCE-style losses derived in this section (Table 2) provide theoretically grounded gradient estimators for the regularized objectives, practical implementations often benefit significantly from stabilization techniques common in policy gradient methods. These techniques aim to reduce variance and control the magnitude of policy updates, which is especially crucial in the off-policy setting where importance weights $w(x)$ and can exacerbate instability.

- **Baseline Subtraction and Regularized Advantage Definition:** This is a standard variance reduction technique. Critically, when combining with stabilization like PPO clipping in this REINFORCE-style context, the term playing the role of the advantage ($\widehat{A}_t$) that gets clipped should

ideally incorporate not just the baselined reward but also the regularization terms derived from the objective's gradient.

Recall the REINFORCE-style gradient structure $\nabla_\theta J(\theta) = \mathbb{E}_{x \sim \pi_{\text{sampling}}}[\text{Weight}(x, \theta) \nabla_\theta \log \pi_\theta(x)]$. The PPO objective involves terms like $w_t \widehat{A}_t$. To align these, we define the regularized advantage $\widehat{A}_t$ such that $w_t \widehat{A}_t$ approximates the key part of $\text{Weight}(x, \theta)$. For example:

- For RKL (Proposition F.4), $\text{Weight}_{\text{RKL}} = w(x)(R(x) - \beta(\log w(x) + 1))$. We define the regularized advantage as $\widehat{A}_t^{\text{RKL}} = (R(x) - b(x)) - \beta(\log w(x) + 1)$.
- For URKL (Proposition F.5), $\text{Weight}_{\text{URKL}} = Z_{\text{old}} w(x)(R(x) - \beta \log w(x))$. Ignoring $Z_{\text{old}}$, we define $\widehat{A}_t^{\text{URKL}} = (R(x) - b(x)) - \beta \log w(x)$.
- For FKL or UFKL, the structure might not cleanly separate into $w(x) \times (\dots)$. In such cases, a common simplification is to use $\widehat{A}_t = R(x) - b(x)$ and accept that the clipping primarily stabilizes the reward term's contribution.

This calculated $\widehat{A}_t$ (incorporating reward, baseline, and KL terms) is then treated as constant using the stop-gradient operator, $\text{SG}(\widehat{A}_t)$, when plugged into the clipping loss function.

- **RPG-Style Objective Clipping (Dual-Clip Variant):** PPO (Schulman et al., 2017) introduces objective clipping to limit the impact of large importance ratios $w(x)$. The Dual-Clip variant (Ye et al., 2020) refines this, particularly for negative advantages, using a lower bound parameter $c > 1$. When applied in the REINFORCE-style setting, the PPO Dual-Clip objective aims to maximize (simplified notation, expectation over $t \sim \pi_{\text{old}}$):

$$J^{\text{DualClip}}(\theta) = \mathbb{E}_t[L_t^{\text{DualClip}}(\theta)]$$

where $\widehat{A}_t$ is the regularized advantage defined above (incorporating $R_t$, $b_t$, and KL terms), $w_t(\theta) = \frac{\pi_\theta(a_t|s_t)}{\pi_{\text{old}}(a_t|s_t)}$, and $L_t^{\text{DualClip}}(\theta)$ is defined based on the sign of $\text{SG}(\widehat{A}_t)$:

- If $\text{SG}(\widehat{A}_t) \geq 0$: $L_t^{\text{DualClip}}(\theta) = \min(w_t(\theta)\,\text{SG}(\widehat{A}_t), \text{clip}(w_t(\theta), 1 - \epsilon_1, 1 + \epsilon_2)\,\text{SG}(\widehat{A}_t))$
- If $\text{SG}(\widehat{A}_t) < 0$: $L_t^{\text{DualClip}}(\theta) = \max(\min(w_t(\theta)\,\text{SG}(\widehat{A}_t), \text{clip}(w_t(\theta), 1 - \epsilon_1, 1 + \epsilon_2)\,\text{SG}(\widehat{A}_t)), c\,\text{SG}(\widehat{A}_t))$

Here, $\epsilon_1, \epsilon_2$ are clipping hyperparameters, and $c$ is the lower bound factor. Note that $\theta$ influences this objective only through $w_t(\theta)$, as $\widehat{A}_t$ is detached via SG.

To implement this using gradient descent (minimizing a loss), we minimize the negative of the PPO Dual-Clip objective. The loss function becomes $\mathcal{L}^{\text{DualClip}}(\theta) = \mathbb{E}_t[\mathcal{L}_t^{\text{DualClip}}(\theta)]$, where $\mathcal{L}_t^{\text{DualClip}}(\theta) = -L_t^{\text{DualClip}}(\theta)$. Explicitly:

- If $\text{SG}(\widehat{A}_t) \geq 0$: $\mathcal{L}_t^{\text{DualClip}}(\theta) = \max(-w_t(\theta)\,\text{SG}(\widehat{A}_t), -\text{clip}(w_t(\theta), 1 - \epsilon_1, 1 + \epsilon_2)\,\text{SG}(\widehat{A}_t))$.
- If $\text{SG}(\widehat{A}_t) < 0$: Let $L_{\text{clip}} = \max(-w_t(\theta)\,\text{SG}(\widehat{A}_t), -\text{clip}(w_t(\theta), 1 - \epsilon_1, 1 + \epsilon_2)\,\text{SG}(\widehat{A}_t))$. Then, $\mathcal{L}_t^{\text{DualClip}}(\theta) = \min(L_{\text{clip}}, -c\,\text{SG}(\widehat{A}_t))$.

This PPO Dual-Clip loss function $\mathcal{L}^{\text{DualClip}}(\theta)$ replaces the simpler REINFORCE-style losses derived earlier (like $\mathcal{L}_{\text{RKL}}^{\text{REINFORCE-style}}$ in Eq. (F.2)). The gradient $\nabla_\theta \mathcal{L}^{\text{DualClip}}(\theta)$ is computed via automatic differentiation, where the gradient flows through $w_t(\theta)$ but is stopped at $\widehat{A}_t$. This approach uses the PPO objective structure with the appropriately defined regularized advantage for stabilization in an off-policy REINFORCE-style update. Algorithm 2 details this implementation.

## H   DETAILED EXPERIMENTAL SETUP

**Hyperparameters.** Unless otherwise specified, all experiments use AdamW optimizer (Loshchilov and Hutter, 2019) with a learning rate of $1 \times 10^{-6}$, a weight decay of 0.1, and gradient clipping at 1.0. Training proceeds for 400 steps, including an initial 10 warm-up steps, after which a constant learning rate is maintained. The global training batch size is 512. For each sample in the batch, we roll out 16 responses using a temperature of 1.0. The per-GPU mini-batch size is 32, and experiments are conducted on 8 NVIDIA H100 GPUs. The maximum training and rollout length is set to 4,096 tokens for a 2K context length and 8,192 tokens for a 4K context length, with dynamic batching enabled. The KL regularization coefficient $\beta$ is set to $1 \times 10^{-4}$.

**Specific Clipping Parameters and Adopted Techniques.** As mentioned in Section 5, we set $(\epsilon_1, \epsilon_2) = (0.2, 0.28)$ for RPG, RPG-REINFORCE and DAPO. For GRPO, we use $(\epsilon_1, \epsilon_2) = (0.1, 0.1)$.

---

**Algorithm 2** REINFORCE-Style RPG with Dual-Clip Stabilization

---

**Require:** Reference policy $\pi_{\text{old}}$, Reward function $R(x)$, Initial policy parameters $\theta_0$
**Require:** KL Component function Compute_KL_Component$(x, \theta, \pi_{\text{old}})$, KL Component Coefficient $\beta$
**Require:** Learning rate $\alpha > 0$, Batch size $N > 0$, Number of epochs $K \geq 1$ per iteration
**Require:** Dual Clip parameters: $\epsilon_1 > 0$ (low), $\epsilon_2 > 0$ (high), $c > 1$
**Require:** Baseline method (e.g., batch average, value function $V_\phi$)
1: Initialize policy parameters $\theta \leftarrow \theta_0$
2: Initialize value function parameters $\phi$ (if baseline uses $V_\phi$)
3: **for** each training iteration **do**
4:     Sample batch $\mathcal{D} = \{x_i\}_{i=1}^N \sim \pi_{\text{old}}$
5:     Compute rewards $R_i$ for $i = 1..N$
6:     Compute baselines $b_i$ for $i = 1..N$ (e.g., $b_i = \frac{1}{N} \sum_j R_j$ or $b_i = V_\phi(x_i)$)
7:     **for** $k = 1$ to $K$ **do**                    ▷ Multiple optimization epochs on the same batch
8:         Initialize batch loss $\mathcal{L}_{\text{batch}} = 0$
9:         **for** $i = 1$ to $N$ **do**
10:             $w_i = \frac{\pi_\theta(x_i)}{\pi_{\text{old}}(x_i)}$                    ▷ Importance weight
11:             $\ell_i = -\log \pi_\theta(x_i)$                    ▷ Negative log probability
12:             $A_{R,i} = R_i - b_i$                    ▷ Baseline-subtracted reward
13:             $C_{\text{KL},i} = \beta \cdot \text{Compute\_KL\_Component}(x_i, \theta, \pi_{\text{old}}(x_i))$                    ▷ KL component
14:             $A'_i = A_{R,i} + \text{SG}(C_{\text{KL},i})/\text{SG}(w_i)$                    ▷ Effective advantage
15:             $\psi_i = A'_i \times \ell_i$                    ▷ Branching term
16:             **if** $\psi_i \geq 0$ **then**
17:                 $w_{\text{high}} = 1 + \epsilon_2$
18:                 **if** $w_i < w_{\text{high}}$ **then**
19:                     $\mathcal{L}_i = \psi_i \times \text{SG}(w_i)$                    ▷ Grad exists
20:                 **else**                    ▷ $w_i \geq w_{\text{high}}$
21:                     $A'_{\text{high}} = A_{R,i} + \text{SG}(C_{\text{KL},i})/\text{SG}(w_{\text{high}})$
22:                     $\psi_{\text{high}} = A'_{\text{high}} \times \text{SG}(\ell_i)$
23:                     $\mathcal{L}_i = \psi_{\text{high}} \times \text{SG}(w_{\text{high}})$
24:                 **end if**
25:             **else**                    ▷ $\psi_i \leq 0$
26:                 $w_{\text{low}} = 1 - \epsilon_1$
27:                 **if** $w_i \leq w_{\text{low}}$ **then**
28:                     $A'_{\text{low}} = A_{R,i} + \text{SG}(C_{\text{KL},i})/\text{SG}(w_{\text{low}})$
29:                     $\psi_{\text{low}} = A'_{\text{low}} \times \text{SG}(\ell_i)$
30:                     $\mathcal{L}_i = \psi_{\text{low}} \times \text{SG}(w_{\text{low}})$
31:                 **else if** $w_i < c$ **then**
32:                     $\mathcal{L}_i = \psi_i \times \text{SG}(w_i)$                    ▷ Grad exists
33:                 **else**                    ▷ $w_i \geq c$
34:                     $\mathcal{L}_i = A_{R,i} \times \text{SG}(\ell_i) \times c + \text{SG}(C_{\text{KL},i}) \times \text{SG}(\ell_i)$
35:                 **end if**
36:             **end if**
37:             $\mathcal{L}_{\text{batch}} = \mathcal{L}_{\text{batch}} + \mathcal{L}_i$
38:         **end for**
39:         $\mathcal{L}(\theta) = \frac{1}{N}\mathcal{L}_{\text{batch}}$                    ▷ Compute average batch loss
40:         $g \leftarrow \nabla_\theta \mathcal{L}(\theta)$                    ▷ Compute gradient
41:         $\theta \leftarrow \text{OptimizerUpdate}(\theta, g, \alpha)$                    ▷ Update policy parameters
42:         **if** using a learned baseline $V_\phi$ **then**
43:             Update value function parameters $\phi$
44:         **end if**
45:     **end for**
46: **end for**
47: **return** Optimized policy parameters $\theta$

---

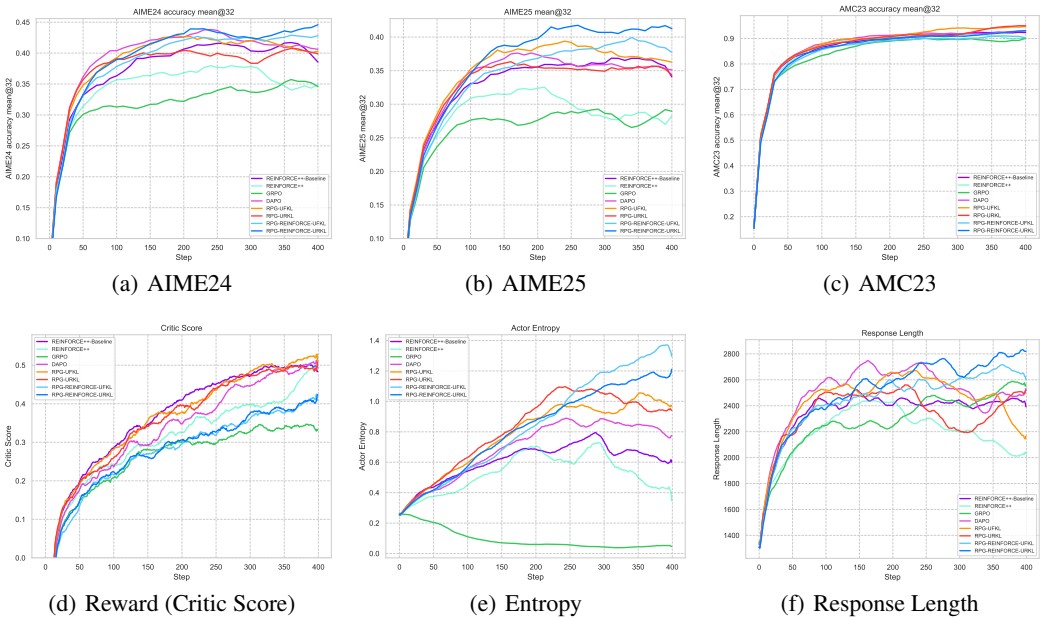

Figure 5: Performance of RPG and REINFORCE-Style Regularized Policy Gradient (RPG-REINFORCE) methods compared to baselines with 4k context length.

# I   ADDITIONAL EXPERIMENT RESULTS

## I.1   THE PERFORMANCE WITH 2K AND 4K CONTEXT LENGTH

The quantitative results in Table 5 demonstrate the competitive performance of the proposed RPG and REINFORCE-style RPG frameworks with 4k context length. On AIME24, RPG-REINFORCE variants lead, with RPG-REINFORCE-URKL achieving the best "Best" score (0.4531) and the best "Last" score (0.4458), while RPG-REINFORCE-UFKL attain a second best "Last" score (0.4281). For AIME25, RPG-REINFORCE-URKL still achieves the top "Best" score (0.4313) and a strong "Last" score (0.4125) and RPG-REINFORCE-UFKL is second only to that. Overall, RPG and RPG-REINFORCE methods rank at or near the top across benchmarks and metrics, while exhibiting stable training dynamics.

Similarly, Table 6 shows the experiment results with 2k context length. It can be observed that RPG and RPG-REINFORCE variants demonstrate robust performance, often competitive with or exceeding baselines. For example, RPG-REINFORCE-UFKL achieves the top "Best" scores for AIME24 (0.3625) and AIME25 (0.3083), and the top "Last" score of AIME25 (0.2927), while RPG-UFKL attains the top "Last" score of AIME24 (0.3427) and the second highest "Last" score of AIME25 (0.2833). Their training curves in Figure 6 generally indicate good stability and effective learning. The consistently high performance across various RPG formulations underscores the utility of the systematically derived KL-regularized objectives explored in this work.

## I.2   ABLATION STUDY

To further investigate our algorithms, we implement an ablation study on the clip ratio and the effect of the KL regularization coefficient.

### I.2.1   ABLATION ON CLIP RATIO

We first implement experiments with different clip ratios on REINFORCE-style RPG algorithms. We choose $(0.1, 0.1)$ and $(0.2, 0.28)$ for $(\epsilon_1, \epsilon_2)$ since they are 2 typical choices of clip ratios (Schulman et al., 2017; Yu et al., 2025), and the performance curves as well as key training dynamics are displayed in Figure 8. It can be observed that although the critic score and response length are similar for different settings, the actor entropy shows a huge difference in trend, demonstrating that an

Table 5: Combined performance metrics with 4k context length on the AIME24, AIME25 and AMC23 mathematical reasoning benchmarks, showing "Last" and "Best" scores. The "Last" score is from the 400th training step, assuming the training process remained stable to that point. The highest score in each column is **bolded**, and the second highest is underlined. RPG and RPG-REINFORCE methods are highlighted with light cyan and light green backgrounds, respectively.

| Method | AIME24 | | AIME25 | | AMC23 | |
|---|---|---|---|---|---|---|
| | **Last** | **Best** | **Last** | **Best** | **Last** | **Best** |
| REINFORCE++-Baseline | 0.3854 | 0.4302 | 0.3406 | 0.3844 | 0.9234 | 0.9328 |
| REINFORCE++ | 0.3490 | 0.3885 | 0.2822 | 0.3479 | 0.8977 | 0.9297 |
| GRPO | 0.3458 | 0.3677 | 0.2896 | 0.3042 | 0.9016 | 0.9109 |
| DAPO | 0.4063 | 0.4479 | 0.3510 | 0.3938 | 0.9297 | 0.9297 |
| RPG-UFKL | 0.4031 | 0.4396 | 0.3625 | 0.3979 | 0.9477 | 0.9500 |
| RPG-URKL | 0.3990 | 0.4219 | 0.3438 | 0.3792 | **0.9500** | **0.9531** |
| RPG-REINFORCE-UFKL | 0.4281 | 0.4375 | 0.3771 | 0.4042 | 0.9023 | 0.9133 |
| RPG-REINFORCE-URKL | **0.4458** | **0.4531** | **0.4125** | **0.4313** | 0.9313 | 0.9352 |

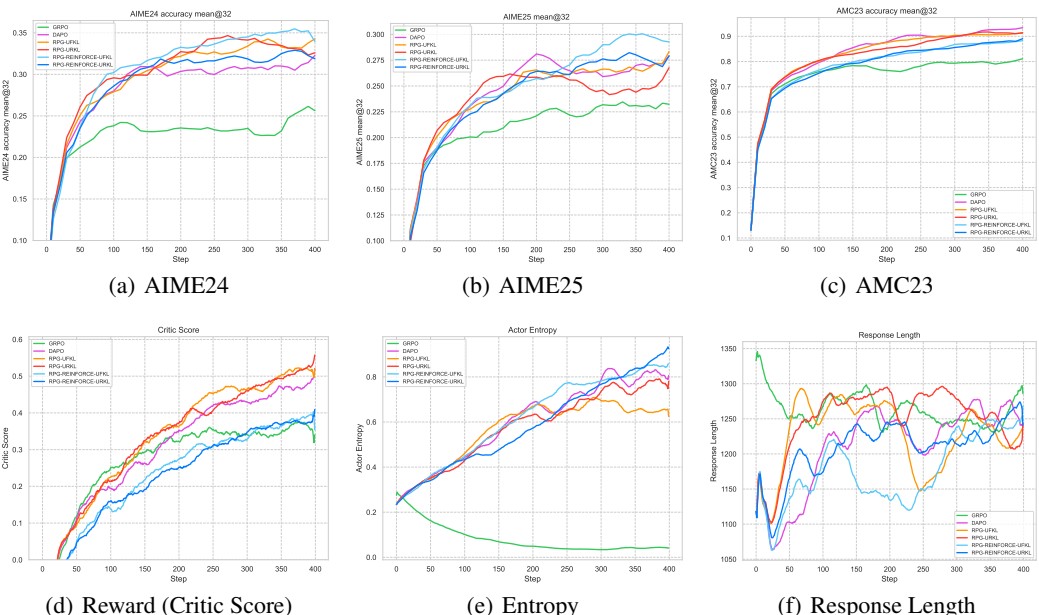

(a) AIME24     (b) AIME25     (c) AMC23

(d) Reward (Critic Score)     (e) Entropy     (f) Response Length

Figure 6: Training dynamics and benchmark performance for fully differentiable Regularized Policy Gradient (RPG) and REINFORCE-Style RPG (RPG-REINFORCE) compared to baselines (GRPO and DAPO) with 2k context length.

adequately higher and clip-higher strategy proposed by DAPO may greatly contribute to the increase of performance by increasing the actor entropy.

### I.2.2 ABLATION ON KL REGULARIZATION COEFFICIENT

We also implement ablation studies on the effect of the KL regularization coefficient. We implement experiments with REINFORCE-style RPG-UFKL (RPG-REINFORCE-UFKL) with $\beta = 1 \times 10^{-3}$ and $1 \times 10^{-4}$, and the results are shown in Figure 9. Figures 9(a) and 9(b) show that the coefficient $1 \times 10^{-4}$ performs better than $1 \times 10^{-3}$, and the trend in response length conforms to the performance, indicating that longer response length may help with the performance improvement.

We also dig into the effect of the iteratively updated reference model. We implement another experiment with no iteratively updated reference model, and display the performance and dynamics

Table 6: Combined performance metrics with 2K context length on the AIME24, and AIME25 mathematical reasoning benchmarks, showing "Last" and "Best" scores. The "Last" score is from the 400th training step, assuming the training process remained stable to that point. The highest score in each column is **bolded**, and the second highest is underlined. RPG and RPG-REINFORCE methods are highlighted with light cyan and light green backgrounds, respectively.

| Method | AIME24 | | AIME25 | |
|---|---|---|---|---|
| | Last | Best | Last | Best |
| GRPO | 0.2563 | 0.2708 | 0.2323 | 0.2479 |
| DAPO | 0.3229 | 0.3281 | 0.2792 | 0.2844 |
| RPG-UFKL | **0.3427** | 0.3479 | 0.2833 | 0.2833 |
| RPG-URKL | 0.3260 | 0.3594 | 0.2677 | 0.2677 |
| RPG-REINFORCE-UFKL | 0.3396 | **0.3625** | **0.2927** | **0.3083** |
| RPG-REINFORCE-URKL | 0.3188 | 0.3417 | 0.2792 | 0.2938 |

Table 7: Combined performance metrics with 4k context length on the Minerva-Math and Olympiad-Bench mathematical reasoning benchmarks, showing "Last" and "Best" scores. The "Last" score is from the 400th training step, assuming the training process remained stable to that point. The highest score in each column is **bolded**, and the second highest is underlined. RPG and RPG-REINFORCE methods are highlighted with light cyan and light green backgrounds, respectively.

| Method | Minerva-Math | | OlympiadBench | |
|---|---|---|---|---|
| | Last | Best | Last | Best |
| REINFORCE++-Baseline | 0.1250 | 0.1471 | 0.4926 | 0.5875 |
| REINFORCE++ | **0.1691** | 0.1728 | 0.4778 | **0.6202** |
| GRPO | 0.1029 | 0.1177 | 0.4926 | 0.5594 |
| DAPO | 0.0993 | 0.1507 | **0.5163** | 0.5727 |
| RPG-UFKL | 0.1397 | **0.2059** | 0.4688 | 0.5564 |
| RPG-URKL | 0.1654 | 0.1654 | 0.4837 | 0.5801 |
| RPG-REINFORCE-UFKL | 0.1360 | 0.1434 | 0.5074 | 0.5564 |
| RPG-REINFORCE-URKL | 0.1140 | 0.1434 | 0.4674 | 0.5816 |

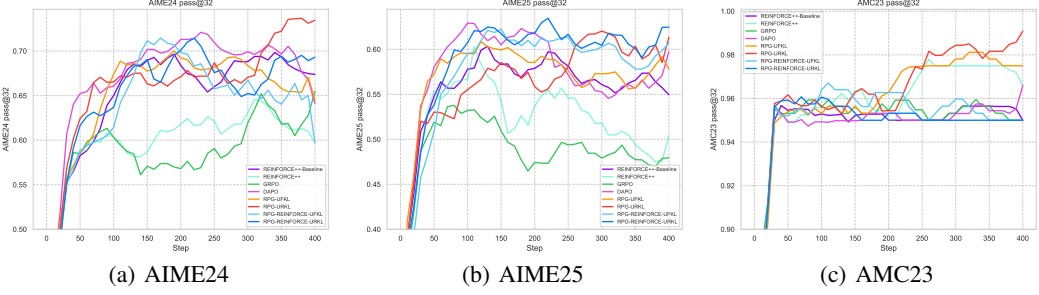

(a) AIME24     (b) AIME25     (c) AMC23

Figure 7: Pass@32 performances for fully differentiable Regularized Policy Gradient (RPG) and REINFORCE-Style RPG (RPG-REINFORCE) compared to baselines with 4k context length.

in Figure 9. It can be observed that the performance recovers with longer response length and much lower actor entropy, showing that longer response length can be an important factor and indicator of the performance on benchmarks.

## I.3 EXPERIMENTS ON QWEN-2.5-7B-INSTRUCT

### I.3.1 REGULARIZED POLICY GRADIENT USING QWEN-2.5-7B-INSTRUCT

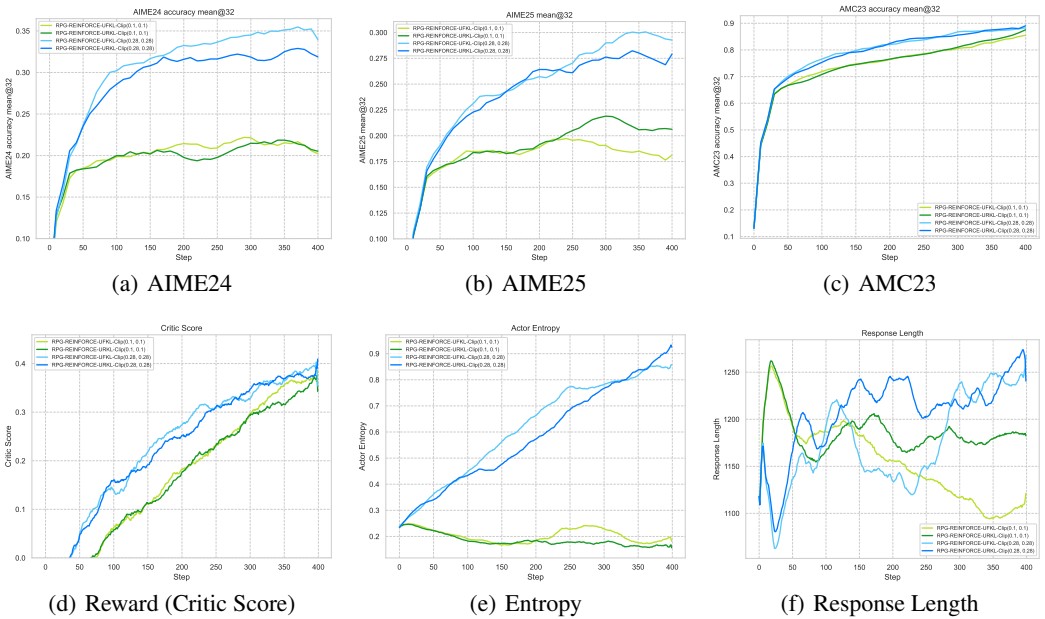

Figure 8: Performance of REINFORCE-Style Regularized Policy Gradient (RPG-REINFORCE) methods with different clip ratios with 2k context length. Plots display accuracy on mathematical reasoning benchmarks (AIME24, AIME25) and key training dynamics (reward, policy entropy, response length).

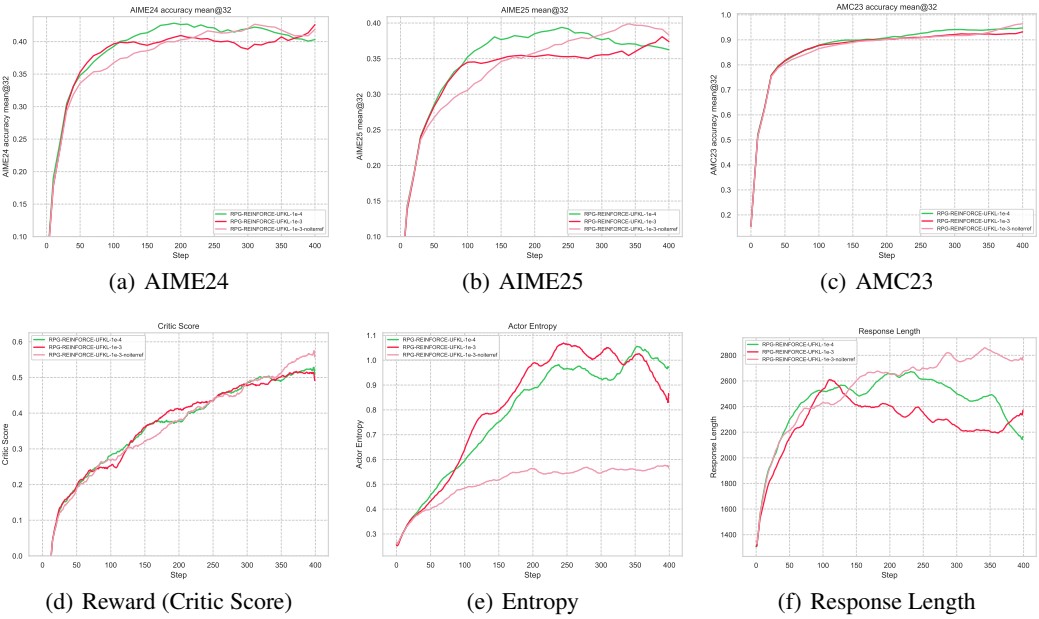

Figure 9: Performance of REINFORCE-Style Regularized Policy Gradient (RPG-REINFORCE) methods with different KL coefficients with 4K context length. Here, "noiterref" indicates the model is trained with no iteratively updated reference model. Plots display accuracy on mathematical reasoning benchmarks (AIME24, AIME25) and key training dynamics (reward, policy entropy, response length).

Table 8: Combined performance metrics on the AMC23, AIME24, and AIME25 mathematical reasoning benchmarks with Qwen-2.5-7B-Instruct model, showing "Last" and "Best" scores. The "Last" score is from the 400th training step, assuming the training process remained stable to that point. The highest score in each column is **bolded**, and the second highest is underlined. RPG and RPG-REINFORCE methods are highlighted with light cyan and light green backgrounds, respectively.

| Method | AMC23 | | AIME24 | | AIME25 | |
|---|---|---|---|---|---|---|
| | Last | Best | Last | Best | Last | Best |
| GRPO | 0.6266 | 0.7250 | 0.1094 | 0.1406 | 0.0281 | 0.0948 |
| REINFORCE++ | 0.7625 | 0.7664 | 0.0521 | 0.1177 | 0.0302 | 0.0740 |
| REINFORCE++-Baseline | 0.8711 | 0.8711 | 0.0990 | 0.1510 | 0.0656 | 0.0969 |
| DAPO | 0.8039 | 0.8734 | 0.0760 | 0.1240 | 0.0531 | 0.1063 |
| RPG-FKL | 0.8695 | **0.8836** | 0.1083 | 0.1490 | 0.0427 | 0.1083 |
| RPG-RKL | 0.8648 | 0.8672 | 0.1167 | 0.1469 | 0.0677 | **0.1240** |
| RPG-UFKL | 0.8703 | 0.8703 | 0.0885 | 0.1427 | **0.0927** | 0.1177 |
| RPG-URKL | 0.8258 | 0.8641 | 0.0875 | 0.1271 | 0.0677 | 0.0917 |
| RPG-REINFORCE-FKL | **0.8727** | 0.8727 | 0.1208 | **0.1667** | 0.0573 | 0.0875 |
| RPG-REINFORCE-RKL | 0.8305 | 0.8516 | 0.1125 | 0.1375 | 0.0490 | 0.0875 |
| RPG-REINFORCE-UFKL | 0.8391 | 0.8602 | **0.1229** | 0.1458 | 0.0740 | 0.0979 |
| RPG-REINFORCE-URKL | 0.8531 | 0.8531 | 0.1208 | 0.1500 | 0.0813 | 0.0938 |

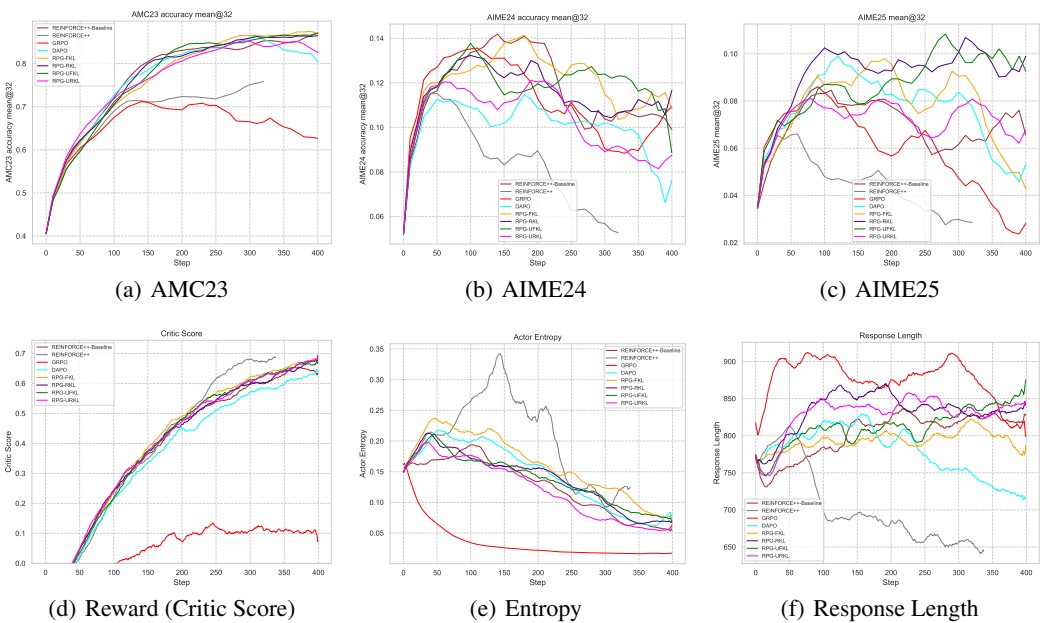

(a) AMC23    (b) AIME24    (c) AIME25

(d) Reward (Critic Score)    (e) Entropy    (f) Response Length

Figure 10: Performance of fully differentiable Regularized Policy Gradient (RPG) methods compared to baselines when using base model: Qwen-2.5-7B-Instruct. Plots display accuracy on mathematical reasoning benchmarks (AMC23, AIME24, AIME25) and key training dynamics (reward, policy entropy, response length).

## J    PROOF OF THEOREM 2.1 (GENERALIZED POLICY GRADIENT THEOREM)

*Proof.* The proof relies on the log-derivative trick, $\nabla_\theta \pi_\theta(x) = \pi_\theta(x) \nabla_\theta \log \pi_\theta(x)$, and the product rule under the integral sign:

$$
\begin{aligned}
\nabla_\theta \mathbb{E}_{x \sim \pi_\theta}[f(x, \theta)] &= \nabla_\theta \int \pi_\theta(x) f(x, \theta) dx \\
&= \int \nabla_\theta(\pi_\theta(x) f(x, \theta)) dx \quad \left(\text{Swap } \nabla, \int\right) \\
&= \int \left((\nabla_\theta \pi_\theta(x)) f(x, \theta) + \pi_\theta(x)(\nabla_\theta f(x, \theta))\right) dx \\
&= \int \left(\pi_\theta(x)(\nabla_\theta \log \pi_\theta(x)) f(x, \theta) + \pi_\theta(x)(\nabla_\theta f(x, \theta))\right) dx \quad \text{(Log-derivative)} \\
&= \int \pi_\theta(x) \left(f(x, \theta) \nabla_\theta \log \pi_\theta(x) + \nabla_\theta f(x, \theta)\right) dx \\
&= \mathbb{E}_{x \sim \pi_\theta}\left[f(x, \theta) \nabla_\theta \log \pi_\theta(x) + \nabla_\theta f(x, \theta)\right].
\end{aligned}
$$

$\square$

## K    PROOFS FOR REGULARIZED POLICY GRADIENTS

This section provides detailed proofs for the theorems presented in Section 3, demonstrating that the gradients of the proposed fully differentiable off-policy surrogate losses correspond to the negative gradients of the respective original objectives. The core tool used is the policy gradient theorem: $\nabla_\theta \mathbb{E}_{x \sim \pi_\theta}[f(x, \theta)] = \mathbb{E}_{x \sim \pi_\theta}[f(x, \theta) \nabla_\theta \log \pi_\theta(x) + \nabla_\theta f(x, \theta)]$. We use the notation $w(x) = \pi_\theta(x) / \pi_{\text{old}}(x)$ for the importance weight.

### K.1    PROOF OF PROPOSITION E.1 (POLICY GRADIENT AND DIFFERENTIABLE LOSS FOR NORMALIZED FORWARD KL)

*Proof.* We start by rewriting the objective function $J_{\text{FKL}}(\theta)$ using expectations with respect to the fixed reference policy $\pi_{\text{old}}$. The first term, the expected reward under $\pi_\theta$, can be rewritten using importance sampling:

$$
\mathbb{E}_{x \sim \pi_\theta}[R(x)] = \int \pi_\theta(x) R(x) dx = \int \frac{\pi_\theta(x)}{\pi_{\text{old}}(x)} \pi_{\text{old}}(x) R(x) dx = \mathbb{E}_{x \sim \pi_{\text{old}}}[w(x) R(x)].
$$

The second term is the forward KL divergence:

$$
\begin{aligned}
\text{KL}(\pi_{\text{old}} \,\|\, \pi_\theta) &= \mathbb{E}_{x \sim \pi_{\text{old}}}\left[\log \frac{\pi_{\text{old}}(x)}{\pi_\theta(x)}\right] \\
&= \mathbb{E}_{x \sim \pi_{\text{old}}}[\log \pi_{\text{old}}(x) - \log \pi_\theta(x)] \\
&= \mathbb{E}_{x \sim \pi_{\text{old}}}[-\log \pi_\theta(x)] + \mathbb{E}_{x \sim \pi_{\text{old}}}[\log \pi_{\text{old}}(x)].
\end{aligned}
$$

Substituting these into the objective function:

$$
\begin{aligned}
J_{\text{FKL}}(\theta) &= \mathbb{E}_{x \sim \pi_{\text{old}}}[w(x) R(x)] - \beta \left(\mathbb{E}_{x \sim \pi_{\text{old}}}[-\log \pi_\theta(x)] + \mathbb{E}_{x \sim \pi_{\text{old}}}[\log \pi_{\text{old}}(x)]\right) \\
&= \mathbb{E}_{x \sim \pi_{\text{old}}}[w(x) R(x) + \beta \log \pi_\theta(x)] - \beta \mathbb{E}_{x \sim \pi_{\text{old}}}[\log \pi_{\text{old}}(x)].
\end{aligned}
$$

Since $\pi_{\text{old}}(x)$ does not depend on $\theta$, the term $\beta \mathbb{E}_{x \sim \pi_{\text{old}}}[\log \pi_{\text{old}}(x)]$ is a constant with respect to $\theta$. Now we compute the gradient $\nabla_\theta J_{\text{FKL}}(\theta)$. Assuming we can swap gradient and expectation (standard assumption in policy gradient methods):

$$
\begin{aligned}
\nabla_\theta J_{\text{FKL}}(\theta) &= \nabla_\theta \mathbb{E}_{x \sim \pi_{\text{old}}}[w(x) R(x) + \beta \log \pi_\theta(x)] \\
&= \mathbb{E}_{x \sim \pi_{\text{old}}}[\nabla_\theta(w(x) R(x) + \beta \log \pi_\theta(x))] \\
&= \mathbb{E}_{x \sim \pi_{\text{old}}}[(\nabla_\theta w(x)) R(x) + \beta \nabla_\theta \log \pi_\theta(x)].
\end{aligned}
$$

We use the identity for the gradient of the importance weight:

$$\nabla_\theta w(x) = \nabla_\theta \left( \frac{\pi_\theta(x)}{\pi_{\text{old}}(x)} \right)$$

$$= \frac{1}{\pi_{\text{old}}(x)} \nabla_\theta \pi_\theta(x)$$

$$= \frac{\pi_\theta(x)}{\pi_{\text{old}}(x)} \frac{\nabla_\theta \pi_\theta(x)}{\pi_\theta(x)}$$

$$= w(x) \nabla_\theta \log \pi_\theta(x).$$

Substituting this back into the gradient expression:

$$\nabla_\theta J_{\text{FKL}}(\theta) = \mathbb{E}_{x \sim \pi_{\text{old}}} \left[ w(x)(\nabla_\theta \log \pi_\theta(x))R(x) + \beta \nabla_\theta \log \pi_\theta(x) \right]$$

$$= \mathbb{E}_{x \sim \pi_{\text{old}}} \left[ \left( w(x)R(x) + \beta \right) \nabla_\theta \log \pi_\theta(x) \right].$$

This proves the first part of the theorem.

Now, consider the surrogate loss function:

$$\mathcal{L}_{\text{FKL}}(\theta) = \mathbb{E}_{x \sim \pi_{\text{old}}} \left[ -w(x)R(x) - \beta \log \pi_\theta(x) \right].$$

We compute its gradient:

$$\nabla_\theta \mathcal{L}_{\text{FKL}}(\theta) = \nabla_\theta \mathbb{E}_{x \sim \pi_{\text{old}}} \left[ -w(x)R(x) - \beta \log \pi_\theta(x) \right]$$

$$= \mathbb{E}_{x \sim \pi_{\text{old}}} \left[ \nabla_\theta(-w(x)R(x) - \beta \log \pi_\theta(x)) \right]$$

$$= \mathbb{E}_{x \sim \pi_{\text{old}}} \left[ -(\nabla_\theta w(x))R(x) - \beta \nabla_\theta \log \pi_\theta(x) \right]$$

$$= \mathbb{E}_{x \sim \pi_{\text{old}}} \left[ -w(x)(\nabla_\theta \log \pi_\theta(x))R(x) - \beta \nabla_\theta \log \pi_\theta(x) \right]$$

$$= -\mathbb{E}_{x \sim \pi_{\text{old}}} \left[ \left( w(x)R(x) + \beta \right) \nabla_\theta \log \pi_\theta(x) \right].$$

Comparing this with the gradient of the objective function, we see that $\nabla_\theta \mathcal{L}_{\text{FKL}}(\theta) = -\nabla_\theta J_{\text{FKL}}(\theta)$. This confirms that minimizing $\mathcal{L}_{\text{FKL}}(\theta)$ corresponds to maximizing $J_{\text{FKL}}(\theta)$ using gradient-based methods. $\square$

### K.2 PROOF OF PROPOSITION 3.2 (POLICY GRADIENT AND DIFFERENTIABLE LOSS FOR UNNORMALIZED FORWARD KL)

*Proof.* We start by expressing the components of $J_{\text{UFKL}}(\theta)$ using expectations over the normalized reference distribution $\widetilde{\pi}_{\text{old}}(x) = \pi_{\text{old}}(x)/Z_{\text{old}}$. The importance weight is $w(x) = \pi_\theta(x)/\pi_{\text{old}}(x)$, which implies $\pi_\theta(x) = w(x)\pi_{\text{old}}(x) = w(x)Z_{\text{old}}\widetilde{\pi}_{\text{old}}(x)$.

The expected reward term:

$$\mathbb{E}_{x \sim \pi_\theta}[R(x)] = \int \pi_\theta(x)R(x)dx = \int w(x)\pi_{\text{old}}(x)R(x)dx$$

$$= \int w(x)Z_{\text{old}}\widetilde{\pi}_{\text{old}}(x)R(x)dx = Z_{\text{old}}\mathbb{E}_{x \sim \widetilde{\pi}_{\text{old}}}[w(x)R(x)].$$

The unnormalized KL divergence term $\text{UKL}(\pi_{\text{old}} \| \pi_\theta)$ has two parts. Part 1 (Generalized KL):

$$\int \pi_{\text{old}}(x) \log \frac{\pi_{\text{old}}(x)}{\pi_\theta(x)} \, dx = \int Z_{\text{old}}\widetilde{\pi}_{\text{old}}(x) \log \frac{\pi_{\text{old}}(x)}{\pi_\theta(x)} \, dx$$

$$= Z_{\text{old}}\mathbb{E}_{x \sim \widetilde{\pi}_{\text{old}}} \left[ \log \frac{1}{w(x)} \right] = Z_{\text{old}}\mathbb{E}_{x \sim \widetilde{\pi}_{\text{old}}} \left[ -\log w(x) \right].$$

Part 2 (Mass Correction):

$$\int (\pi_\theta(x) - \pi_{\text{old}}(x))dx = \int (w(x)\pi_{\text{old}}(x) - \pi_{\text{old}}(x))dx$$

$$= \int (w(x) - 1)\pi_{\text{old}}(x)dx = \int (w(x) - 1)Z_{\text{old}}\widetilde{\pi}_{\text{old}}(x)dx$$

$$= Z_{\text{old}}\mathbb{E}_{x \sim \widetilde{\pi}_{\text{old}}}[w(x) - 1] = Z_{\text{old}}\mathbb{E}_{x \sim \widetilde{\pi}_{\text{old}}}[w(x)] - Z_{\text{old}}.$$

Combining these parts for the UKL term:

$$\text{UKL}(\pi_{\text{old}} \| \pi_\theta) = Z_{\text{old}} \mathbb{E}_{x \sim \widetilde{\pi}_{\text{old}}} \left[ -\log w(x) \right] + Z_{\text{old}} \mathbb{E}_{x \sim \widetilde{\pi}_{\text{old}}} [w(x)] - Z_{\text{old}}.$$

Now, substitute everything into the objective $J_{\text{UFKL}}(\theta)$:

$$J_{\text{UFKL}}(\theta) = Z_{\text{old}} \mathbb{E}_{x \sim \widetilde{\pi}_{\text{old}}} [w(x)R(x)] - \beta \left( Z_{\text{old}} \mathbb{E}_{x \sim \widetilde{\pi}_{\text{old}}} [-\log w(x)] + Z_{\text{old}} \mathbb{E}_{x \sim \widetilde{\pi}_{\text{old}}} [w(x)] - Z_{\text{old}} \right)$$
$$= Z_{\text{old}} \mathbb{E}_{x \sim \widetilde{\pi}_{\text{old}}} [w(x)R(x) + \beta \log w(x) - \beta w(x) + \beta].$$

To compute the gradient $\nabla_\theta J_{\text{UFKL}}(\theta)$, we differentiate the terms inside the expectation. The constant term $\beta Z_{\text{old}}$ (arising from $\beta$ inside the expectation) vanishes upon differentiation.

$$\nabla_\theta J_{\text{UFKL}}(\theta) = \nabla_\theta \left( Z_{\text{old}} \mathbb{E}_{x \sim \widetilde{\pi}_{\text{old}}} [w(x)R(x) + \beta \log w(x) - \beta w(x)] \right)$$
$$= Z_{\text{old}} \mathbb{E}_{x \sim \widetilde{\pi}_{\text{old}}} [\nabla_\theta(w(x)R(x)) + \beta \nabla_\theta(\log w(x)) - \beta \nabla_\theta(w(x))].$$

We need the gradients of $w(x)$ and $\log w(x)$:

$$\nabla_\theta w(x) = w(x) \nabla_\theta \log \pi_\theta(x) \quad \text{(as derived in Proposition E.1 proof)}$$
$$\nabla_\theta \log w(x) = \nabla_\theta(\log \pi_\theta(x) - \log \pi_{\text{old}}(x)) = \nabla_\theta \log \pi_\theta(x).$$

Substituting these into the gradient expression:

$$\nabla_\theta J_{\text{UFKL}}(\theta) = Z_{\text{old}} \mathbb{E}_{x \sim \widetilde{\pi}_{\text{old}}} [(\nabla_\theta w(x))R(x) + \beta \nabla_\theta \log \pi_\theta(x) - \beta(\nabla_\theta w(x))]$$
$$= Z_{\text{old}} \mathbb{E}_{x \sim \widetilde{\pi}_{\text{old}}} [w(x)R(x) \nabla_\theta \log \pi_\theta(x) + \beta \nabla_\theta \log \pi_\theta(x) - \beta w(x) \nabla_\theta \log \pi_\theta(x)]$$
$$= Z_{\text{old}} \mathbb{E}_{x \sim \widetilde{\pi}_{\text{old}}} [(w(x)R(x) - \beta w(x) + \beta) \nabla_\theta \log \pi_\theta(x)]$$
$$= Z_{\text{old}} \mathbb{E}_{x \sim \widetilde{\pi}_{\text{old}}} \left[ \left( w(x)R(x) - \beta(w(x) - 1) \right) \nabla_\theta \log \pi_\theta(x) \right].$$

This proves the first part of the theorem.

Now, consider the surrogate loss function:

$$\mathcal{L}_{\text{UFKL}}(\theta) = Z_{\text{old}} \mathbb{E}_{x \sim \widetilde{\pi}_{\text{old}}} \left[ -w(x)R(x) + \beta \left( w(x) - \log w(x) - 1 \right) \right].$$

We compute its gradient:

$$\nabla_\theta \mathcal{L}_{\text{UFKL}}(\theta) = Z_{\text{old}} \mathbb{E}_{x \sim \widetilde{\pi}_{\text{old}}} [\nabla_\theta(-w(x)R(x)) + \beta \nabla_\theta(w(x) - \log w(x) - 1)]$$
$$= Z_{\text{old}} \mathbb{E}_{x \sim \widetilde{\pi}_{\text{old}}} [-(\nabla_\theta w(x))R(x) + \beta(\nabla_\theta w(x) - \nabla_\theta \log w(x))]$$
$$= Z_{\text{old}} \mathbb{E}_{x \sim \widetilde{\pi}_{\text{old}}} [-w(x)R(x) \nabla_\theta \log \pi_\theta(x) + \beta(w(x) \nabla_\theta \log \pi_\theta(x) - \nabla_\theta \log \pi_\theta(x))]$$
$$= Z_{\text{old}} \mathbb{E}_{x \sim \widetilde{\pi}_{\text{old}}} \left[ \left( -w(x)R(x) + \beta w(x) - \beta \right) \nabla_\theta \log \pi_\theta(x) \right]$$
$$= -Z_{\text{old}} \mathbb{E}_{x \sim \widetilde{\pi}_{\text{old}}} \left[ \left( w(x)R(x) - \beta(w(x) - 1) \right) \nabla_\theta \log \pi_\theta(x) \right].$$

Comparing this with the gradient of the objective function, we find $\nabla_\theta \mathcal{L}_{\text{UFKL}}(\theta) = -\nabla_\theta J_{\text{UFKL}}(\theta)$. This confirms the surrogate loss function. Note that the constant $-1$ inside the logarithm term in the loss $\mathcal{L}_{\text{UFKL}}$ corresponds to the constant $\beta Z_{\text{old}}$ in the objective $J_{\text{UFKL}}$ and does not affect the gradient. $\qquad\square$

### K.3 PROOF OF PROPOSITION E.3 (POLICY GRADIENT AND DIFFERENTIABLE LOSS FOR NORMALIZED REVERSE KL)

*Proof.* We rewrite the objective function $J_{\text{RKL}}(\theta)$ using expectations with respect to $\pi_{\text{old}}$. The expected reward term is $\mathbb{E}_{x \sim \pi_\theta}[R(x)] = \mathbb{E}_{x \sim \pi_{\text{old}}}[w(x)R(x)]$, as shown previously. The reverse KL divergence term is:

$$\text{KL}(\pi_\theta \| \pi_{\text{old}}) = \mathbb{E}_{x \sim \pi_\theta} \left[ \log \frac{\pi_\theta(x)}{\pi_{\text{old}}(x)} \right]$$
$$= \mathbb{E}_{x \sim \pi_\theta} [\log w(x)]$$
$$= \int \pi_\theta(x) \log w(x) dx$$
$$= \int \frac{\pi_\theta(x)}{\pi_{\text{old}}(x)} \pi_{\text{old}}(x) \log w(x) dx$$
$$= \mathbb{E}_{x \sim \pi_{\text{old}}} [w(x) \log w(x)].$$

Substituting these into the objective function:

$$J_{\text{RKL}}(\theta) = \mathbb{E}_{x \sim \pi_{\text{old}}}[w(x)R(x)] - \beta\mathbb{E}_{x \sim \pi_{\text{old}}}[w(x)\log w(x)] = \mathbb{E}_{x \sim \pi_{\text{old}}}[w(x)R(x) - \beta w(x)\log w(x)].$$

Now we compute the gradient $\nabla_\theta J_{\text{RKL}}(\theta)$:

$$\begin{aligned} \nabla_\theta J_{\text{RKL}}(\theta) &= \nabla_\theta \mathbb{E}_{x \sim \pi_{\text{old}}}\left[w(x)R(x) - \beta w(x)\log w(x)\right] \\ &= \mathbb{E}_{x \sim \pi_{\text{old}}}\left[\nabla_\theta(w(x)R(x)) - \beta\nabla_\theta(w(x)\log w(x))\right]. \end{aligned}$$

We need the gradient of $w(x)\log w(x)$:

$$\begin{aligned} \nabla_\theta(w(x)\log w(x)) &= (\nabla_\theta w(x))\log w(x) + w(x)\nabla_\theta(\log w(x)) \\ &= (w(x)\nabla_\theta \log \pi_\theta(x))\log w(x) + w(x)(\nabla_\theta \log \pi_\theta(x)) \\ &= w(x)\nabla_\theta \log \pi_\theta(x)(\log w(x) + 1). \end{aligned}$$

Substituting this and $\nabla_\theta w(x) = w(x)\nabla_\theta \log \pi_\theta(x)$ into the gradient expression for $J_{\text{RKL}}(\theta)$:

$$\begin{aligned} \nabla_\theta J_{\text{RKL}}(\theta) &= \mathbb{E}_{x \sim \pi_{\text{old}}}\left[(\nabla_\theta w(x))R(x) - \beta w(x)\nabla_\theta \log \pi_\theta(x)(\log w(x) + 1)\right] \\ &= \mathbb{E}_{x \sim \pi_{\text{old}}}\left[w(x)(\nabla_\theta \log \pi_\theta(x))R(x) - \beta w(x)(\log w(x) + 1)\nabla_\theta \log \pi_\theta(x)\right] \\ &= \mathbb{E}_{x \sim \pi_{\text{old}}}\left[w(x)\Big(R(x) - \beta(\log w(x) + 1)\Big)\nabla_\theta \log \pi_\theta(x)\right]. \end{aligned}$$

This proves the first part of the theorem.

Now, consider the surrogate loss function:

$$\mathcal{L}_{\text{RKL}}(\theta) = \mathbb{E}_{x \sim \pi_{\text{old}}}\left[w(x)\big(-R(x) + \beta \log w(x)\big)\right].$$

We compute its gradient:

$$\begin{aligned} \nabla_\theta \mathcal{L}_{\text{RKL}}(\theta) &= \nabla_\theta \mathbb{E}_{x \sim \pi_{\text{old}}}\left[-w(x)R(x) + \beta w(x)\log w(x)\right] \\ &= \mathbb{E}_{x \sim \pi_{\text{old}}}\left[\nabla_\theta(-w(x)R(x)) + \beta\nabla_\theta(w(x)\log w(x))\right] \\ &= \mathbb{E}_{x \sim \pi_{\text{old}}}\left[-(\nabla_\theta w(x))R(x) + \beta w(x)\nabla_\theta \log \pi_\theta(x)(\log w(x) + 1)\right] \\ &= \mathbb{E}_{x \sim \pi_{\text{old}}}\left[-w(x)(\nabla_\theta \log \pi_\theta(x))R(x) + \beta w(x)(\log w(x) + 1)\nabla_\theta \log \pi_\theta(x)\right] \\ &= \mathbb{E}_{x \sim \pi_{\text{old}}}\left[w(x)\Big(-R(x) + \beta(\log w(x) + 1)\Big)\nabla_\theta \log \pi_\theta(x)\right] \\ &= -\mathbb{E}_{x \sim \pi_{\text{old}}}\left[w(x)\Big(R(x) - \beta(\log w(x) + 1)\Big)\nabla_\theta \log \pi_\theta(x)\right]. \end{aligned}$$

Comparing this with the gradient of the objective function, we confirm that $\nabla_\theta \mathcal{L}_{\text{RKL}}(\theta) = -\nabla_\theta J_{\text{RKL}}(\theta)$. □

### K.4 PROOF OF PROPOSITION 3.6 (POLICY GRADIENT AND DIFFERENTIABLE LOSS FOR UNNORMALIZED REVERSE KL)

*Proof.* We again express the objective components using expectations over the normalized reference distribution $\widetilde{\pi}_{\text{old}}(x) = \pi_{\text{old}}(x)/Z_{\text{old}}$, with $w(x) = \pi_\theta(x)/\pi_{\text{old}}(x)$.

The expected reward term: $\mathbb{E}_{x \sim \pi_\theta}[R(x)] = Z_{\text{old}}\mathbb{E}_{x \sim \widetilde{\pi}_{\text{old}}}[w(x)R(x)]$.

The unnormalized reverse KL divergence $\text{UKL}(\pi_\theta \| \pi_{\text{old}})$ has two parts. Part 1 (Generalized KL):

$$\begin{aligned} \int \pi_\theta(x)\log \frac{\pi_\theta(x)}{\pi_{\text{old}}(x)}\, dx &= \int \pi_\theta(x)\log w(x)dx \\ &= \int w(x)\pi_{\text{old}}(x)\log w(x)dx \\ &= \int w(x)Z_{\text{old}}\widetilde{\pi}_{\text{old}}(x)\log w(x)dx \\ &= Z_{\text{old}}\mathbb{E}_{x \sim \widetilde{\pi}_{\text{old}}}[w(x)\log w(x)]. \end{aligned}$$

Part 2 (Mass Correction):

$$\int (\pi_{\text{old}}(x) - \pi_\theta(x))dx = \int \pi_{\text{old}}(x)dx - \int \pi_\theta(x)dx$$

$$= Z_{\text{old}} - \int w(x)\pi_{\text{old}}(x)dx$$

$$= Z_{\text{old}} - \int w(x)Z_{\text{old}}\widetilde{\pi}_{\text{old}}(x)dx$$

$$= Z_{\text{old}} - Z_{\text{old}}\mathbb{E}_{x\sim\widetilde{\pi}_{\text{old}}}[w(x)].$$

Combining these for the UKL term:

$$\text{UKL}(\pi_\theta\|\pi_{\text{old}}) = Z_{\text{old}}\mathbb{E}_{x\sim\widetilde{\pi}_{\text{old}}}[w(x)\log w(x)] + Z_{\text{old}} - Z_{\text{old}}\mathbb{E}_{x\sim\widetilde{\pi}_{\text{old}}}[w(x)].$$

Now, substitute into the objective $J_{\text{URKL}}(\theta)$:

$$J_{\text{URKL}}(\theta) = Z_{\text{old}}\mathbb{E}_{x\sim\widetilde{\pi}_{\text{old}}}[w(x)R(x)] - \beta\left(Z_{\text{old}}\mathbb{E}_{x\sim\widetilde{\pi}_{\text{old}}}[w(x)\log w(x)] + Z_{\text{old}} - Z_{\text{old}}\mathbb{E}_{x\sim\widetilde{\pi}_{\text{old}}}[w(x)]\right)$$

$$= Z_{\text{old}}\mathbb{E}_{x\sim\widetilde{\pi}_{\text{old}}}\left[w(x)R(x) - \beta w(x)\log w(x) - \beta + \beta w(x)\right].$$

We compute the gradient $\nabla_\theta J_{\text{URKL}}(\theta)$. The constant term $-\beta Z_{\text{old}}$ vanishes upon differentiation.

$$\nabla_\theta J_{\text{URKL}}(\theta) = \nabla_\theta\left(Z_{\text{old}}\mathbb{E}_{x\sim\widetilde{\pi}_{\text{old}}}[w(x)R(x) - \beta w(x)\log w(x) + \beta w(x)]\right)$$

$$= Z_{\text{old}}\mathbb{E}_{x\sim\widetilde{\pi}_{\text{old}}}\left[\nabla_\theta(w(x)R(x)) - \beta\nabla_\theta(w(x)\log w(x)) + \beta\nabla_\theta w(x)\right].$$

Using the previously derived gradients $\nabla_\theta w(x) = w(x)\nabla_\theta\log\pi_\theta(x)$ and $\nabla_\theta(w(x)\log w(x)) = w(x)\nabla_\theta\log\pi_\theta(x)(\log w(x) + 1)$:

$$\nabla_\theta J_{\text{URKL}}(\theta) = \nabla_\theta\left(Z_{\text{old}}\mathbb{E}_{x\sim\widetilde{\pi}_{\text{old}}}[w(x)R(x) - \beta w(x)\log w(x) + \beta w(x)]\right)$$

$$= Z_{\text{old}}\mathbb{E}_{x\sim\widetilde{\pi}_{\text{old}}}\left[\nabla_\theta(w(x)R(x)) - \beta\nabla_\theta(w(x)\log w(x)) + \beta\nabla_\theta w(x)\right]$$

$$= Z_{\text{old}}\mathbb{E}_{x\sim\widetilde{\pi}_{\text{old}}}\left[(\nabla_\theta w(x))R(x) - \beta w(x)\nabla_\theta\log\pi_\theta(x)(\log w(x) + 1) + \beta(\nabla_\theta w(x))\right]$$

$$= Z_{\text{old}}\mathbb{E}_{x\sim\widetilde{\pi}_{\text{old}}}[w(x)R(x)\nabla_\theta\log\pi_\theta(x) - \beta w(x)(\log w(x) + 1)\nabla_\theta\log\pi_\theta(x)$$

$$+\beta w(x)\nabla_\theta\log\pi_\theta(x)]$$

$$= Z_{\text{old}}\mathbb{E}_{x\sim\widetilde{\pi}_{\text{old}}}\left[w(x)\nabla_\theta\log\pi_\theta(x)\Big(R(x) - \beta(\log w(x) + 1) + \beta\Big)\right]$$

$$= Z_{\text{old}}\mathbb{E}_{x\sim\widetilde{\pi}_{\text{old}}}\left[w(x)\nabla_\theta\log\pi_\theta(x)\Big(R(x) - \beta\log w(x)\Big)\right]$$

$$= Z_{\text{old}}\mathbb{E}_{x\sim\widetilde{\pi}_{\text{old}}}\left[w(x)\Big(R(x) - \beta\log w(x)\Big)\nabla_\theta\log\pi_\theta(x)\right].$$

This proves the first part of the theorem.

Now, consider the surrogate loss function:

$$\mathcal{L}_{\text{URKL}}(\theta) = Z_{\text{old}}\mathbb{E}_{x\sim\widetilde{\pi}_{\text{old}}}\left[-w(x)R(x) + \beta\big(w(x)\log w(x) - w(x)\big)\right].$$

We compute its gradient:

$$\nabla_\theta\mathcal{L}_{\text{URKL}}(\theta) = Z_{\text{old}}\mathbb{E}_{x\sim\widetilde{\pi}_{\text{old}}}\left[\nabla_\theta(-w(x)R(x)) + \beta\nabla_\theta(w(x)\log w(x) - w(x))\right]$$

$$= Z_{\text{old}}\mathbb{E}_{x\sim\widetilde{\pi}_{\text{old}}}\left[-(\nabla_\theta w(x))R(x) + \beta(\nabla_\theta(w(x)\log w(x)) - \nabla_\theta w(x))\right]$$

$$= Z_{\text{old}}\mathbb{E}_{x\sim\widetilde{\pi}_{\text{old}}}[-w(x)R(x)\nabla_\theta\log\pi_\theta(x)$$

$$+\beta\big(w(x)(\log w(x) + 1)\nabla_\theta\log\pi_\theta(x) - w(x)\nabla_\theta\log\pi_\theta(x)\big)]$$

$$= Z_{\text{old}}\mathbb{E}_{x\sim\widetilde{\pi}_{\text{old}}}[-w(x)R(x)\nabla_\theta\log\pi_\theta(x) + \beta w(x)\log w(x)\nabla_\theta\log\pi_\theta(x)]$$

$$= Z_{\text{old}}\mathbb{E}_{x\sim\widetilde{\pi}_{\text{old}}}\left[w(x)\Big(-R(x) + \beta\log w(x)\Big)\nabla_\theta\log\pi_\theta(x)\right]$$

$$= -Z_{\text{old}}\mathbb{E}_{x\sim\widetilde{\pi}_{\text{old}}}\left[w(x)\Big(R(x) - \beta\log w(x)\Big)\nabla_\theta\log\pi_\theta(x)\right].$$

Comparing this with the gradient of the objective function, we confirm that $\nabla_\theta\mathcal{L}_{\text{URKL}}(\theta) = -\nabla_\theta J_{\text{URKL}}(\theta)$. The constant term $+1$ (corresponding to $-\beta Z_{\text{old}}$ in the objective) that appeared in the derivation in Section 3.2 does not affect the gradient and is often omitted from the final loss expression used in practice. $\qquad\square$

## L    PROOFS FOR REINFORCE-STYLE REGULARIZED POLICY GRADIENTS

This section provides justifications for the REINFORCE-style surrogate loss functions presented in Section 4 (Theorems F.1 to F.5). These proofs demonstrate how automatic differentiation applied to the proposed losses, utilizing the stop-gradient operator SG, yields the correct gradient direction (negative of the objective gradient derived in Section 3).

The core idea relies on the operational definition of the stop-gradient operator $\mathrm{SG}(\cdot)$ within automatic differentiation frameworks: $\nabla_\theta \mathrm{SG}(f(\theta)) = 0$, while the forward computation uses the value of $f(\theta)$. We use the notation $w(x) = \pi_\theta(x)/\pi_{\mathrm{old}}(x)$.

### L.1    PROOF OF PROPOSITION F.1 (REINFORCE-STYLE POLICY GRADIENT FOR FORWARD KL)

*Proof.* The objective is $J_{\mathrm{FKL}}(\theta) = \mathbb{E}_{\pi_\theta}[R(x)] - \beta \,\mathrm{KL}(\pi_{\mathrm{old}} \,\|\, \pi_\theta)$. From Proposition E.1, its gradient is:

$$\nabla_\theta J_{\mathrm{FKL}}(\theta) = \mathbb{E}_{x \sim \pi_{\mathrm{old}}}\left[\underbrace{(w(x)R(x) + \beta)}_{\mathrm{Weight}_{\mathrm{FKL}}(x,\theta)} \nabla_\theta \log \pi_\theta(x)\right].$$

The proposed REINFORCE-style surrogate loss is:

$$\mathcal{L}_{\mathrm{FKL}}^{\mathrm{REINFORCE\text{-}style}}(\theta) = -\mathbb{E}_{x \sim \pi_{\mathrm{old}}}\left[\mathrm{SG}\left(w(x)R(x) + \beta\right) \log \pi_\theta(x)\right].$$

We compute the gradient of this loss as it would be computed by an automatic differentiation system. Assuming the gradient can be swapped with the expectation:

$$\nabla_\theta \mathcal{L}_{\mathrm{FKL}}^{\mathrm{REINFORCE\text{-}style}}(\theta) = -\mathbb{E}_{x \sim \pi_{\mathrm{old}}}\left[\nabla_\theta \left(\mathrm{SG}\left(w(x)R(x) + \beta\right) \log \pi_\theta(x)\right)\right]$$

$$= -\mathbb{E}_{x \sim \pi_{\mathrm{old}}}\left[\underbrace{\left(\nabla_\theta \mathrm{SG}\left(w(x)R(x) + \beta\right)\right)}_{=0 \text{ by definition of SG}} \log \pi_\theta(x)\right.$$

$$\left. + \,\mathrm{SG}\left(w(x)R(x) + \beta\right)\left(\nabla_\theta \log \pi_\theta(x)\right)\right]$$

$$= -\mathbb{E}_{x \sim \pi_{\mathrm{old}}}\left[\mathrm{SG}\left(w(x)R(x) + \beta\right) \nabla_\theta \log \pi_\theta(x)\right].$$

This gradient expression, when used in an optimization algorithm (where SG is conceptually removed), corresponds to applying updates proportional to:

$$-\left(-\mathbb{E}_{x \sim \pi_{\mathrm{old}}}\left[(w(x)R(x) + \beta) \nabla_\theta \log \pi_\theta(x)\right]\right) = \nabla_\theta J_{\mathrm{FKL}}(\theta).$$

Thus, minimizing $\mathcal{L}_{\mathrm{FKL}}^{\mathrm{REINFORCE\text{-}style}}(\theta)$ using gradient descent with automatic differentiation effectively performs gradient ascent on the original objective $J_{\mathrm{FKL}}(\theta)$.    $\square$

### L.2    PROOF OF PROPOSITION F.3 ((REINFORCE-STYLE POLICY GRADIENT FOR UNNORMALIZED FORWARD KL)

*Proof.* The objective is $J_{\mathrm{UFKL}}(\theta) = \mathbb{E}_{\pi_\theta}[R(x)] - \beta \,\mathrm{UKL}(\pi_{\mathrm{old}} \| \pi_\theta)$. From Proposition 3.2, its gradient is:

$$\nabla_\theta J_{\mathrm{UFKL}}(\theta) = \mathbb{E}_{x \sim \widetilde{\pi}_{\mathrm{old}}}\left[\underbrace{Z_{\mathrm{old}}\left(w(x)R(x) - \beta\left(w(x) - 1\right)\right)}_{\mathrm{Weight}_{\mathrm{UFKL}}(x,\theta)} \nabla_\theta \log \pi_\theta(x)\right].$$

The proposed REINFORCE-style surrogate loss is:

$$\mathcal{L}_{\mathrm{UFKL}}^{\mathrm{REINFORCE\text{-}style}}(\theta) = -\mathbb{E}_{x \sim \widetilde{\pi}_{\mathrm{old}}}\left[\mathrm{SG}\left(Z_{\mathrm{old}}\left(w(x)R(x) - \beta(w(x) - 1)\right)\right) \log \pi_\theta(x)\right].$$

Computing the gradient via automatic differentiation:

$$\nabla_\theta \mathcal{L}_{\text{UFKL}}^{\text{REINFORCE-style}}(\theta) = -\mathbb{E}_{x \sim \widetilde{\pi}_{\text{old}}} \left[ \nabla_\theta \left( \text{SG} \left( Z_{\text{old}}(\dots) \right) \log \pi_\theta(x) \right) \right]$$

$$= -\mathbb{E}_{x \sim \widetilde{\pi}_{\text{old}}} \left[ \underbrace{\left( \nabla_\theta \, \text{SG}(Z_{\text{old}}(\dots)) \right)}_{=0} \log \pi_\theta(x) + \text{SG}(Z_{\text{old}}(\dots))(\nabla_\theta \log \pi_\theta(x)) \right]$$

$$= -\mathbb{E}_{x \sim \widetilde{\pi}_{\text{old}}} \left[ \text{SG} \left( Z_{\text{old}} \left( w(x)R(x) - \beta(w(x) - 1) \right) \right) \nabla_\theta \log \pi_\theta(x) \right].$$

This gradient corresponds to the update direction $-\nabla_\theta J_{\text{UFKL}}(\theta)$ when the SG is dropped. Minimizing this loss achieves gradient ascent on $J_{\text{UFKL}}(\theta)$. If $Z_{\text{old}}$ is omitted, the same argument applies to the proportionally scaled objective and loss. □

### L.3 PROOF OF PROPOSITION F.4 (REINFORCE-STYLE LOSS)

*Proof.* The objective is $J_{\text{RKL}}(\theta) = \mathbb{E}_{\pi_\theta}[R(x)] - \beta \, \text{KL}(\pi_\theta \| \pi_{\text{old}})$. From Proposition E.3, its gradient is:

$$\nabla_\theta J_{\text{RKL}}(\theta) = \mathbb{E}_{x \sim \pi_{\text{old}}} \left[ \underbrace{w(x) \Big( R(x) - \beta(\log w(x) + 1) \Big)}_{\text{Weight}_{\text{RKL}}(x, \theta)} \nabla_\theta \log \pi_\theta(x) \right].$$

The proposed REINFORCE-style surrogate loss is:

$$\mathcal{L}_{\text{RKL}}^{\text{REINFORCE-style}}(\theta) = -\mathbb{E}_{x \sim \pi_{\text{old}}} \left[ \text{SG} \left( w(x) \left( R(x) - \beta \log w(x) - \beta \right) \right) \log \pi_\theta(x) \right].$$

Computing the gradient via automatic differentiation:

$$\nabla_\theta \mathcal{L}_{\text{RKL}}^{\text{REINFORCE-style}}(\theta) = -\mathbb{E}_{x \sim \pi_{\text{old}}} \left[ \nabla_\theta \left( \text{SG} \left( w(x)(\dots) \right) \log \pi_\theta(x) \right) \right]$$

$$= -\mathbb{E}_{x \sim \pi_{\text{old}}} \left[ \underbrace{\left( \nabla_\theta \, \text{SG}(w(x)(\dots)) \right)}_{=0} \log \pi_\theta(x) + \text{SG}(w(x)(\dots))(\nabla_\theta \log \pi_\theta(x)) \right]$$

$$= -\mathbb{E}_{x \sim \pi_{\text{old}}} \left[ \text{SG} \left( w(x) \left( R(x) - \beta \log w(x) - \beta \right) \right) \nabla_\theta \log \pi_\theta(x) \right].$$

This gradient corresponds to the update direction $-\nabla_\theta J_{\text{RKL}}(\theta)$ when the SG is dropped. Minimizing this loss achieves gradient ascent on $J_{\text{RKL}}(\theta)$. □

### L.4 PROOF OF PROPOSITION F.5 (REINFORCE-STYLE LOSS FOR UNNORMALIZED REVERSE KL)

*Proof.* The objective is $J_{\text{URKL}}(\theta) = \mathbb{E}_{\pi_\theta}[R(x)] - \beta \, \text{UKL}(\pi_\theta \| \pi_{\text{old}})$. From Proposition 3.6, its gradient is:

$$\nabla_\theta J_{\text{URKL}}(\theta) = \mathbb{E}_{x \sim \widetilde{\pi}_{\text{old}}} \left[ \underbrace{Z_{\text{old}} w(x) \Big( R(x) - \beta \log w(x) \Big)}_{\text{Weight}_{\text{URKL}}(x, \theta)} \nabla_\theta \log \pi_\theta(x) \right].$$

The proposed REINFORCE-style surrogate loss is:

$$\mathcal{L}_{\text{URKL}}^{\text{REINFORCE-style}}(\theta) = -\mathbb{E}_{x \sim \widetilde{\pi}_{\text{old}}} \left[ \text{SG} \left( Z_{\text{old}} w(x) \left( R(x) - \beta \log w(x) \right) \right) \log \pi_\theta(x) \right].$$

Computing the gradient via automatic differentiation:

$$\nabla_\theta \mathcal{L}_{\text{URKL}}^{\text{REINFORCE-style}}(\theta) = -\mathbb{E}_{x \sim \widetilde{\pi}_{\text{old}}} \left[ \nabla_\theta \left( \text{SG} \left( Z_{\text{old}} w(x)(\dots) \right) \log \pi_\theta(x) \right) \right]$$

$$= -\mathbb{E}_{x \sim \widetilde{\pi}_{\text{old}}} \left[ \underbrace{\left( \nabla_\theta \, \text{SG}(Z_{\text{old}} w(x)(\dots)) \right)}_{=0} \log \pi_\theta(x) \right.$$

$$\left. + \text{SG}(Z_{\text{old}} w(x)(\dots))(\nabla_\theta \log \pi_\theta(x)) \right]$$

$$= -\mathbb{E}_{x \sim \widetilde{\pi}_{\text{old}}} \left[ \text{SG} \left( Z_{\text{old}} w(x) \left( R(x) - \beta \log w(x) \right) \right) \nabla_\theta \log \pi_\theta(x) \right].$$

This gradient corresponds to the update direction $-\nabla_\theta J_{\text{URKL}}(\theta)$ when the SG is dropped. Minimizing this loss achieves gradient ascent on $J_{\text{URKL}}(\theta)$. If $Z_{\text{old}}$ is omitted, the same argument applies to the proportionally scaled objective and loss. $\square$

