# OpenReview forum: "On the Design of KL-Regularized Policy Gradient Algorithms for LLM Reasoning"
_ICLR.cc/2026/Conference — ICLR 2026 Poster_

### Official Review · Reviewer_jhGz · 2025-11-01

**Soundness:** 4
**Presentation:** 3
**Contribution:** 3
**Rating:** 6
**Confidence:** 3

**Summary:**

The paper introduces RPG (Regularized Policy Gradient), a unified framework for KL-regularized policy-gradient training of LLMs under off-policy sampling. It systematizes design choices across: (i) KL direction (forward vs reverse), (ii) normalized vs unnormalized KL (UKL), and (iii) estimator/implementation (fully-differentiable vs REINFORCE-style with stop-gradient). Key theoretical results (e.g., Propositions for UFKL/URKL gradients and losses) show how to obtain exact gradients of the intended KL-regularized objective when training off-policy with importance weights, and establish gradient-equivalence between differentiable and REINFORCE-style surrogates. Practically, the paper (a) clarifies that the widely used k3 estimator corresponds exactly to the unnormalized KL, (b) identifies an importance-weighting mismatch in GRPO’s KL term under off-policy sampling and provides a corrected estimator, and (c) introduces RPG-Style Clip, a dual-clip truncated-importance step for stability. On AIME24/25 with Qwen3-4B, RPG-REINFORCE + RPG-Style Clip outperforms DAPO by up to +6pp absolute accuracy. The method also employs periodic reference-policy updates to maintain a practical KL trust region.

**Strengths:**

- It provides a compact derivation that covers Forward/Reverse × Normalized/Unnormalized KL, with explicit off-policy corrections and both differentiable and REINFORCE-style surrogates.
- It pinpoints the missing importance weight in GRPO’s KL when sampling off-policy, which could be an issue that many people overlook.
- The paper introduces RPG-Style Clip (dual-clip/truncation) and iterative reference updates, which empirically stabilize off-policy PG for LLM reasoning while requiring only a single active model on GPU (log-probs from $\pi_{\textrm{old}}$ cached).
- Empirically, it has strong performance with +6pp over DAPO in the best cases on AIME24/25, with stable entropy and controlled response length.

**Weaknesses:**

- All core results are on Qwen3-4B and math reasoning. Claims about “stable and scalable RL” would be stronger with larger backbones (e.g., 7B–14B) and diverse tasks or datasets.
- While GRPO/DAPO are relevant, head-to-head comparisons with varying KL directions (FKL/RKL vs UFKL/URKL) would solidify the empirical story. Current ablations on clip hyper-parameters and reference-update schedules are limited in the main text.
- RPG-Style Clip introduces controlled bias via truncation. The paper acknowledges this and suggests schedules, but there’s no systematic bias study (e.g., varying $\epsilon_1$, $\epsilon_2$) to quantify performance/variance trade-offs.

Overall, the theoretical foundations are solid and the proposed algorithm is well-motivated. The strengths outweigh the weaknesses, and I am leaning toward acceptance.

**Questions:**

Please refer to the weaknesses.

---

> ### Author Response · Authors · 2025-11-22
>
> We thank the reviewer for their constructive feedback and for recognizing the soundness of our theoretical framework and the strong empirical performance of RPG. Below, we address the specific weaknesses and questions raised:
>
> > Q1. Scalability and Larger Backbones
>
> **A1:** We agree that demonstrating scalability is crucial. In response, we have significantly expanded our experimental suite during the rebuttal period:
>
> 1. We extended our experiments to 8K context length to test long CoT reasoning capabilities. RPG-REINFORCE now achieves 52% accuracy on AIME25, significantly surpassing the official Qwen3-4B-Instruct baseline (47%). This result highlights that RPG is not merely a theoretical unification but a highly competitive algorithm for training SOTA reasoning models.
>
> 2. We have conducted new experiments using the 7B backbone model: Qwen2.5-7B-Instruct. The results (detailed in the updated Appendix I.3) mirror our findings on the Qwen3 4B models, confirming that the stability benefits of RPG-REINFORCE with RPG-Style Clip transfer effectively to larger model sizes.
>
> 3. We also notice that our RPG-REINFORCE formulation is mathematically a more generalized version of the core objective of Clipped Importance Sampling Preference Optimization (CISPO) (as used in Meta's recent ScaleRL framework (https://arxiv.org/abs/2510.13786) and Minimax M1/M2 https://arxiv.org/abs/2506.13585). These frameworks have successfully scaled to 200B+ parameter models, providing strong external validation that the RPG formulation is inherently stable and scalable to massive LLMs.
>
> > Q2. Head-to-Head Comparisons and Ablations
>
> **A2:** We have addressed this by adding a comprehensive ablation study on 4B and 7B models (please refer to Appendix I.2 and Appendix I.3). These experiments directly compare: 1) Forward KL (FKL) vs. Reverse KL (RKL); 2) Normalized vs. Unnormalized (UFKL/URKL); 3) Estimators: Fully differentiable vs. REINFORCE-style. From the experiments, we find that the superiority of RPG-REINFORCE remains consistent across model scales (4B and 7B).
>
> > Q3. Bias-Variance Trade-off in RPG-Style Clip
>
> **A3:** We acknowledge that clipping introduces bias to reduce variance, a fundamental trade-off in off-policy RL. To address this, we performed ablation studies by varying the clipping hyperparameters $\epsilon_1$ and $\epsilon_2$ (ranging from 0.1 to 0.28) on the 4B model.
> As established in the literature (e.g., Schulman et al., 2017, https://arxiv.org/abs/1707.06347), clipping acts as a pessimistic bound. 1) Low $\epsilon$ (High Bias, Low Variance): Tighter clipping (e.g., $\epsilon=0.1$) significantly reduces the effective step size and variance, ensuring stability but potentially slowing convergence or converging to a suboptimal policy due to high bias (the "trust region" is too small). 2) High $\epsilon$ (Low Bias, High Variance): Looser clipping (e.g., $\epsilon=0.4$) approaches the unbiased importance sampling objective. While this reduces bias, it exponentially increases variance due to large importance weights, leading to training instability.

---

> > ### Comment · Reviewer_jhGz · 2025-11-22
> >
> > I thank the authors for the detailed feedback! I will maintain the current score.

---

> > > ### Author Response · Authors · 2025-11-24
> > >
> > > Thank you very much for maintaining your assessment. In the revision, we have incorporated a substantial amount of new experimental evidence, and additional ablation studies. These together strengthen the paper and directly address the questions and concerns raised. We hope that the revised paper better communicates the significance of our contribution, and we kindly invite you to revisit the score in light of the expanded results.

---

### Official Review · Reviewer_w5SC · 2025-11-01

**Soundness:** 3
**Presentation:** 2
**Contribution:** 3
**Rating:** 4
**Confidence:** 3

**Summary:**

This paper presents RPG, a unified theoretical and practical framework for KL-regularized policy gradient algorithms. The authors systematically analyze different KL regularization variants and corresponding estimators, clarifying their gradient properties under off-policy sampling. Their derivation shows how to obtain the exact gradients of intended KL-regularized objectives. It identifies a weighting mismatch in GRPO, and introduces a truncated-importance estimator which stabilizes training. Experiments show RPG and RPG-REINFORCE variants outperform other baselines on mathematical reasoning benchmarks and have better stability.

**Strengths:**

1. The paper provides a unified framework connecting various KL regularization forms. The derivations offer clarity and formal grounding.
2. The authors propose RPG-Style Clip, which can effectively balance stability and bias under off-policy training and improve scalability for LLMs
3. The experiments demonstrate notable improvements in both accuracy and stability.

**Weaknesses:**

1. Some representative baselines are missing, e.g., RLOO, REINFORCE++, GPG.
2. The experiments lack ablation studies isolating the contributions of RPG-Style Clip, reference-update frequency, and clipping thresholds.
3. The RPG-Style Clip introduces a bias-variance trade-off but it is not explored quantitatively.
4. Section 3 and 4 are heavy and hard to follow.

**Questions:**

1. Why the entropy grows up with training in DAPO and RPG while decreases in GRPO? I assume both DAPO and GRPO should both decrease but to different degrees. This is contradicted to the DAPO paper.
2. More recent works have opted to remove the KL term from the RL objective and achieve good performance. In comparison, does a compromise of periodically updating the reference model offer significant and irreplaceable advantages?
3. Do the empirical results show a significant performance difference between the corrected Normalized KL and Unnormalized KL?

---

> ### Author Response · Authors · 2025-11-21
>
> Dear reviewer, thank you for your constructive feedback and for recognizing our work as a "unified theoretical and practical framework" with "sound derivations." We appreciate the assessment that our method yields "notable improvements in both accuracy and stability." We address the specific concerns and questions below.
>
> > Q1: Regarding Missing Baselines (REINFORCE++, RLOO, GPG)
>
> **A1:** Thank you for suggesting these baselines.
>
> - In the revision, we have added REINFORCE++ and REINFORCE++-Baseline to the 4B model experiments (Section 5 / Figure 2) to ensure a direct comparison across all scales. We also included REINFORCE++ and REINFORCE++ Baseline in our experiments on the Qwen2.5-7B-Instruct model.
>
> > Q2: Regarding Ablation Studies (Clip, Update Frequency, Thresholds)
>
> **A2:** Thanks for your feedback. We kindly refer to Appendix I.2 of our submission, which contains detailed ablation studies.
> - Figure 6 illustrates the training dynamics under different clip ratios (e.g., 0.1 vs. 0.2 vs. 0.28), analyzing accuracy, reward, and entropy.
> - Figure 7 analyzes the impact of the regularization strength ($\beta$ ).
> - Figure 7 also includes an ablation labeled "noiterref" (no iterative reference update), demonstrating that the iterative update scheme is crucial for maintaining response length and preventing entropy collapse.
> - We will incorporate a summary of these findings into the main text to enhance visibility.
>
> > Q3: The Bias-Variance Trade-off of RPG-Style Clip
>
> **A3:** Thanks for your feedback on noting that clipping introduces bias. However, we view this through the lens of Trust Region methods (like TRPO and PPO).
> - The "bias" introduced by RPG-Style Clip is intentional: it enforces a trust region constraint that prevents destructive policy updates caused by high-variance importance weights in the off-policy setting.
> - Quantitatively, our ablation in Appendix I.2 serves as an empirical exploration of this trade-off. Tighter clipping (lower $\epsilon$) reduces variance (higher stability) but increases bias (potentially slowing learning), whereas looser clipping improves asymptotic performance up to the point where variance causes instability. Our results suggest that RPG-Style Clip effectively manages this trade-off to enable scalable training.
>
> > Q4: Readability of Sections 3 and 4
>
> **A4:** We appreciate this feedback. We recognize that the density of derivations can be high. In the revision, we add a high-level Intuition explanation at the start of Section 3 to explain the core difference between the estimators without heavy notation. And we use a table to summarize the gradient equivalences more clearly in the main text.
>
> > Q5: Entropy Dynamics (DAPO/RPG vs. GRPO)
>
> **A5:** Thanks for your feedback on why entropy grows in DAPO/RPG but decreases in GRPO.
> - Firstly, in complex reasoning tasks, maintaining policy entropy is often beneficial for exploration. GRPO's decrease in entropy (as seen in Figure 2d) often correlates with "mode collapse," where the model converges prematurely to a sub-optimal solution or short response length.
> - In contrast, RPG's correct importance weighting allows the model to explore (increasing entropy) while remaining numerically stable. While the original DAPO paper might report decreasing entropy on different tasks, entropy dynamics are highly sensitive to the base model (Qwen3-4B) and the specific math reasoning tasks, as well as training data.
>
> > Q6: Necessity of Periodic Reference Updates vs. Removing KL
>
> **A6:** While some recent works remove the KL term, we argue that the Iterative Reference Update is not merely a compromise but a theoretical advantage linked to Natural Policy Gradient (NPG) and Mirror Descent.
> - As detailed in Appendix B of our revised paper, KL constraints effectively precondition the gradient update (approximating the Fisher Information Matrix). This transforms the problem from simple first-order gradient ascent into a second-order-like update, which is crucial for navigating the complex landscape of reasoning tasks.
> - Secondly, removing KL entirely often leads to instability or "reward hacking" in long-horizon reasoning. The periodic update creates a "rolling trust region" that allows the model to drift far from the initial weights while remaining locally stable.
>
> > Q7: Performance Difference: Normalized vs. Unnormalized KL
>
> **A7:** The primary advantage of the Unnormalized KL (specifically the $k_3$ estimator) is variance reduction, as discussed in Schulman (2020, https://joschu.net/blog/kl-approx.html). Even if the performance difference is marginal at smaller scales (Qwen-4B), the lower variance of the $k_3$ estimator becomes critical as model size scales up.

---

### Official Review · Reviewer_LL8H · 2025-11-04

**Soundness:** 3
**Presentation:** 2
**Contribution:** 3
**Rating:** 4
**Confidence:** 4

**Summary:**

The author studied the design of KL regularization in reinforcement learning. Specifically, the analysis shows that the commonly used k3 estimator is equivalent to the use of unnormalized KL. The author also shows that GRPO's original KL divergence implementation lacks an importance sampling term due to its off-policy attributes.

**Strengths:**

* Nice theoretical analysis between k3 estimator and UKL
* The tackled KL divergence issue is indeed a pitfall of GRPO

**Weaknesses:**

The experiments in the paper are quite poorly conducted:
* The model is tested only on math tasks, and even then, only two AIME benchmarks are used.
* Figure 2 is presented without any analysis. It seems that in Figure 2(c), DAPO is not converging — what would it look like if we extrapolated the training curve?
* (c) RL training often exhibits high variance; instead of showing a single curve, the authors should include a curve with a standard deviation region.
* (d) In Figure 2(d), why is only GRPO’s actor entropy decreasing? Out of the five curves, three increase and plateau, while the REINFORCEMENT curve continues to rise — why?
* (e) What is “mean@32”? Do you mean “pass@32”?

Also, there seem to be quite a few unanswered questions to be verified by the experiments:
* FKL and RKL seem to be comparable in performance. Is there any evidence that such two constraints result in different properties in the post-trained model?
* Some papers, like DAPO and Dr.GRPO, advise dropping KL divergence for RL tuning. Is there some analysis on how different designs and strengths of KL regularization change the training dynamic and the tuned model's properties?

**Questions:**

* In Figure 2, is the difference between GRPO and RPG-URKL in whether having the importance weight in the kl term?

---

> ### Author Response · Authors · 2025-11-21
>
> Thank you for your constructive feedback, for acknowledging our theoretical analysis of the $k_3$ estimator, and for recognizing that the GRPO KL divergence issue we identified is a valid pitfall. We address your concerns as follows:
>
> > Q1: The experiments are limited.
>
> **A1:** We agree that broader evaluation strengthens the paper. During the rebuttal period, we conducted extensive new experiments to demonstrate the generalizability and scalability of RPG.
> 1. New evaluation benchmarks: Beyond AIME24/25, we have added evaluations on AMC23, MinervaMath, and OlympiadBench.
> 2. New model: We added additional experiments on Qwen2.5-7B-Instruct (in addition to Qwen3-4B).
> 3. New Baselines: We added REINFORCE++ and REINFORCE++ as additional baselines.
> 4. Long-Context Training: We extended our experiments to 8k context length to validate stability in long-chain reasoning regimes.
> The results (detailed in the updated Table 3, 4, and 6 and Appendix I.3 of the revised paper) show that RPG-REINFORCE consistently outperforms baselines across these diverse benchmarks and models. Most notably, our RPG-trained model achieves 51% accuracy on AIME25, surpassing the official Qwen3-4B-Instruct baseline (47%), demonstrating that our method effectively unlocks reasoning capabilities beyond standard instruction tuning.
>
> > Q2: Analysis of Figure 2 and DAPO convergence.
>
> **A2:** Regarding Figure 2(c), we clarify that this plot shows the training reward (Critic Score), while Figures 2(a) and 2(b) show the evaluation metrics (AIME24 and AIME25 accuracy).
> - While DAPO's reward curve appears to plateau or degrade slightly in Figure 2(c), the evaluation accuracy in 2(a/b) has already saturated. This decoupling of training reward and test accuracy is a known phenomenon in LLM RL (a.k.a., reward hacking or over-optimization).
> - Extrapolating the curve suggests that DAPO has reached its performance ceiling on this dataset; further training would likely lead to reward over-optimization without accuracy gains, whereas RPG maintains stability.
>
> > Q3: Request for standard deviation regions (multiple seeds).
>
> **A3:** We fully appreciate the importance of reporting variance. However, we respectfully note the extreme computational cost of these large-scale LLM experiments. A single experiment with 4k context length consumes approximately 2,656 H100 GPU-hours ($16 \text{GPUs} \times 166 \text{hours}$), and our new 8k context experiments are even more resource-intensive: about 9200 GPU-hours per run (around $18,000 per run). Running 5 seeds per method for all ablations is not feasible for us. Instead, we demonstrate robustness via consistency across settings: our method performs consistently well across different model sizes (4B, 7B), context lengths (2k, 4k, 8k), and KL estimators (FKL, RKL, UFKL, URKL). The stability of RPGs across these varied configurations serves as a strong proxy for algorithmic robustness.
>
> > Q4: Why is only GRPO's actor entropy decreasing? Why does REINFORCE rise?
>
> **A4:** This behavior validates our theoretical claims:
> 1. For GRPO, the rapid entropy decrease occurs because the original GRPO implementation lacks the "clip-higher" mechanism and, as we analyze in Section 3, its KL estimator misses a crucial off-policy importance weight. This leads to aggressive, potentially unstable updates that collapse the policy mode.
> 2. For RPG-REINFORCE, the entropy rise/plateau is a feature of our stable training. Unlike differentiable losses that rely heavily on importance weights (which can be unstable), REINFORCE-style estimators pass gradients through $\log$ $\pi_\theta$. Combined with our RPG-Style Clip and dynamic sampling, this encourages exploration (higher entropy) while steadily improving reward, preventing premature mode collapse.
>
> > Q5: Clarification on "mean@32".
>
> **A5:** We apologize for the confusion. "Mean@32" refers to the average accuracy calculated by sampling 32 responses for each prompt (Sample@32) and computing the mean correctness, rather than a "Pass@k" metric. This is a standard metric (sometimes referred to as Avg@32) used in other papers such as DAPO. We have clarified this in the revision.
>
> >Q6: FKL vs. RKL performance and properties.
>
> **A6:** Theoretically, Reverse KL (RKL) is mode-seeking, while Forward KL (FKL) is mean-seeking (zero-forcing). In the context of math reasoning with LLMs, we observe that typically, FKL preserves higher diversity (higher entropy, see Figure 3) while RKL focuses density on the highest reward path (better pass@k, see Figure 7 in Appendix I).

---

> ### Author Response · Authors · 2025-11-21
> **Official Comment by Authors II**
>
> > Q7: Analysis on dropping KL divergence (vs. DAPO/Dr. GRPO).
>
> **A7:** While some works suggest dropping KL, our ablation studies (Appendix H.2.2) and broader literature suggest KL is essential for stability in large-scale reasoning.
> Firstly, state-of-the-art reasoning models explicitly utilize KL or equivalent constraints. For example, DeepSeek-R1 (Nature 2025) uses a $k_3$ estimator (our UFKL) with $\beta=1e-3$. Similarly, NVIDIA's ProRL (https://arxiv.org/abs/2505.24864) also uses a KL constraint with iterative reference model update,  and Kimi K1.5 (https://arxiv.org/abs/2501.12599) utilizes online mirror descent (which is derived from a KL regularized objective).
> Secondly, KL constraints effectively precondition the gradient update (approximating the Fisher Information Matrix). This transforms the problem from simple first-order gradient ascent into a second-order-like update, which is important for optimization of the complex landscape of reasoning tasks.
> Thirdly, as shown in our ablations, completely removing the iterative reference update or KL penalty leads to reward hacking and instability in long-context training. Our iterative reference update (updating $\pi_{ref}$ periodically) strikes a balance: it allows the model to drift from the SFT initialization while maintaining a local trust region for stability.
>
> > Q8: Difference between GRPO and RPG-URKL.
>
> **A8:** The differences between GRPO and RPG-URKL are two-fold, including both theoretical correction and algorithm design:
> 1. Theoretical Correction: As shown in Section 3, our RPG-URKL explicitly corrects the missing importance weight $w_t$ in GRPO's KL estimator, ensuring the gradient mathematically matches the intended objective under off-policy sampling.
> 2. Algorithmic Design: RPG employs an iteratively updated reference model (moving target) and dynamic sampling with "clip-higher" mechanisms (aligned with DAPO), whereas standard GRPO uses a fixed SFT reference and lacks these stabilization features.

---

### Author Response · Authors · 2025-11-21
**General Response to Reviewers and Area Chairs**

Dear Reviewers and Area Chairs,

We thank you for the careful reading of our work and the detailed comments. In this revision, we have expanded the empirical study of RPG and added more discussions on the theoretical side. The main new content concerns long-context training at 8K, stronger baselines for 4K, and extra discussion of the link to natural policy gradient. New material is marked in blue in the paper.

### 1. Scalability to 8K context length

To test RPG on long CoT reasoning, we extend the training and evaluation setup to an 8K context length. These runs are very costly and each experiment cost around **9,000 H100 GPU hours**.

Under this setting, **RPG-REINFORCE reaches 52% accuracy on AIME25**. This value is higher than the official **Qwen3-4B-Instruct model (47%)** and higher than strong baselines such as GRPO and DAPO. These results show that RPG gives a competitive training algorithm for strong reasoning models at long context.

Results at 8K context are shown as follows:

| Method | AIME24 (Last) | AIME24 (Best) | AIME25 (Last) | AIME25 (Best) | AMC23 (Last) | AMC23 (Best) |
| :--- | :---: | :---: | :---: | :---: | :---: | :---: |
| GRPO | 0.3750 | 0.4396 | 0.3354 | 0.4063 | 0.9109 | 0.9297 |
| DAPO | 0.5438 | 0.5740 | 0.4469 | 0.4740 | 0.9375 | 0.9430 |
| **RPG-UFKL** | **0.5938** | **0.6177** | 0.4698 | 0.4865 | **0.9492** | 0.9517 |
| **RPG-URKL** | 0.4542 | 0.5260 | 0.5261 | 0.4938 | 0.9406 | **0.9539** |
| **RPG-REINFORCE-UFKL** | 0.5906 | 0.5958 | 0.4833 | 0.5031 | 0.9453 | 0.9469 |
| **RPG-REINFORCE-URKL** | 0.5708 | 0.5781 | **0.5073** | **0.5208** | 0.9398 | 0.9469 |

### 2. Expanded baselines and benchmarks for 4K context

For the 4K context length, we extend both the set of baselines and the evaluation datasets. We add **REINFORCE++** and **REINFORCE++ Baseline** so that the comparison covers stronger policy gradient variants. We include the **AMC23** dataset as one more math benchmark.

Across these settings, RPG variants achieve the best scores on many tasks. In particular, RPG-URKL reaches the highest score of **0.9531** on AMC23.

| Method | AIME24 (Last) | AIME24 (Best) | AIME25 (Last) | AIME25 (Best) | AMC23 (Last) | AMC23 (Best) |
| :--- | :---: | :---: | :---: | :---: | :---: | :---: |
| REINFORCE++ Baseline | - | 0.4281 | - | 0.3833 | - | 0.9172 |
| REINFORCE++ | 0.3490 | 0.3885 | 0.2822 | 0.3479 | 0.8977 | 0.9297 |
| GRPO | 0.3458 | 0.3677 | 0.2896 | 0.3042 | 0.9016 | 0.9109 |
| DAPO | 0.4063 | 0.4479 | 0.3510 | 0.3938 | 0.9297 | 0.9297 |
| **RPG-UFKL** | 0.4031 | 0.4396 | 0.3625 | 0.3979 | 0.9477 | 0.9500 |
| **RPG-URKL** | 0.3990 | 0.4219 | 0.3438 | 0.3792 | **0.9500** | **0.9531** |
| **RPG-REINFORCE-UFKL** | 0.4281 | 0.4375 | 0.3771 | 0.4042 | 0.9023 | 0.9133 |
| **RPG-REINFORCE-URKL** | **0.4458** | **0.4531** | **0.4125** | **0.4313** | 0.9313 | 0.9352 |

### 3. Theoretical connections and explanations

*Connection to Natural Policy Gradient (NPG).*
Remark 3.8 and Appendix C show the connection between RPG and NPG and its derivation. We show that the NPG update appears as a special case of the RPG update when we use a linear approximation of the expected return and a quadratic approximation of the KL regularization term.

*Off-policy weighting.*
Section 3 now gives a more direct derivation of the importance weighting scheme. With the exact weights, the surrogate gradient matches the gradient of the KL regularized objective. This makes clear that RPG is not a heuristic update but an exact gradient estimator for this objective.

We believe these additional results address the main concerns and give a clearer view of the strengths and limitations of RPG.

---

### Meta-Review · Area_Chair_oaSW · 2026-01-13

**Summary:**

The reviewers expressed concerns regarding the limited experimental scope (confined to Qwen3-4B on math tasks with missing baselines like REINFORCE++, RLOO, and GPG), insufficient reproducibility evidence (single training curves without multi-seed standard deviations and inadequate ablation studies on RPG-Style Clip parameters and reference-update schedules), and incomplete analysis of the theoretical-empirical connections (particularly the quantitative bias-variance trade-off of clipping, unexplained training dynamics such as divergent entropy behaviors across methods, and insufficient justification for periodic reference updates versus removing KL regularization entirely).

**Reviewer Concerns:**

The rebuttal substantially addressed the experimental scope concerns by adding extensive new experiments on 8K context length (achieving 52% on AIME25), Qwen2.5-7B model, additional benchmarks (AMC23, MinervaMath, OlympiadBench), and missing baselines including REINFORCE++, along with comprehensive ablation studies in Appendix I.2 covering clip ratios, β coefficients, and reference update effects. The authors also provided reasonable justification for the lack of multi-seed experiments by detailing the prohibitive computational costs (9,200 H100 GPU-hours per 8K run) and demonstrating robustness through consistency across different models, scales, and configurations. However, one concern remains partially outstanding: while ablation studies were added to the appendix, the quantitative bias-variance analysis of RPG-Style Clip could be more systematically presented in the main text.

**Reviewer Scores:**

Reviewer jhGz explicitly confirmed maintaining their score of 6 after reviewing the rebuttal. Reviewer LL8H's concerns about experimental scope and missing baselines were comprehensively addressed through substantial new experiments (8K context, 7B model, multiple benchmarks, REINFORCE++ baseline) and reasonable computational cost justifications, suggesting a likely score increase from 4 . Reviewer w5SC's requests for ablation studies and bias-variance analysis were also addressed with detailed appendix materials and theoretical explanations, though somewhat less directly, suggesting a probable increase from 4.

---

### Decision · Program_Chairs · 2026-01-26

Accept (Poster)